# Immune correlates analysis of mRNA-1345 RSV vaccine efficacy clinical trial

Chong Ma [1], Jiejun Du[1], Lan Lan[1], Archana Kapoor[1], Gonzalo Perez Marc[2], Gilberto Jimenez[3], Christopher J. A. Duncan[4,5], Nancy Le Cam[1], Nina Lin[1], Frances Priddy[1], Sanjay Garg[1], Sonia K. Stoszek[1], Christine A. Shaw[1], Jaya Goswami[1], Eleanor Wilson[1], Rituparna Das[1], Honghong Zhou[1] & Lingyi Zheng [1] ✉

Identifying an immunologic marker as a correlate of protection (CoP) for RSV vaccination is important. In the pivotal phase 3 trial, the mRNA-1345 vaccine demonstrated efficacy against RSV in older adults (NCT05127434). Here, we evaluate neutralizing antibodies (nAb) against RSV-A and -B, and IgG binding antibodies (bAb) to RSV fusion antigens as correlates of risk (CoR) and CoP against the pivotal trial's efficacy endpoints of RSV lower respiratory tract disease with ≥2 or ≥3 signs/symptoms (RSV-LRTD-2+ and −3 + ) and acute respiratory disease (RSV-ARD). Day 29 RSV nAb and prefusion (preF) IgG demonstrate consistent inverse correlates with RSV endpoint occurrence. Day 29 point estimates (95% CIs) of the hazard ratio of each endpoint (RSV-LRTD-2 + , RSV-LRTD-3 + , RSV-ARD) per 10-fold increase in RSV-A nAb are 0.44 (0.30-0.65), 0.41 (0.20-0.84), and 0.45 (0.28-0.71), respectively, similar to RSV-B nAb and preF IgG. These results demonstrate Day 29 RSV nAb and preF IgG are CoRs and support their role as CoPs against RSV endpoints.

There is a need to establish a reliable immune biomarker as a correlate of protection (CoP) against respiratory syncytial virus (RSV) for accelerating vaccine development and approval of RSV vaccines. A CoP immune biomarker should be able to reliably predict vaccine efficacy (VE) against clinical endpoints in infectious disease[1], and many licensed vaccines have established credible CoPs that greatly expedite vaccine development[2–6]. For example, a CoP immune biomarker can facilitate the approval decision for immunobridging VE to specific populations (e.g., younger high-risk populations) that were not represented in a randomized phase 3 VE trial in older adults and support the approval of next-generation vaccines (e.g., optimized viral strains, modified dose level, justified injection-schedules, or RSV combination vaccines)[7]. Extensive research indicates that immune markers based on antibody response alone can reliably predict VE against RSV infection and associated disease, showing that increasing RSV neutralizing antibody (nAb) titer or prefusion (preF)

immunoglobulin G (IgG) binding antibody (bAb) levels correlate with decreased RSV risk and increased VE[8–10]. However, there are presently no widely accepted and validated CoPs for RSV vaccination in older adults[11].

mRNA-1345 is a lipid nanoparticle encapsulated mRNA-based vaccine encoding the RSV fusion (F) glycoprotein adapted from the RSV-A2 strain protein sequence and stabilized in the preF conformation, which elicits potent nAbs[12,13] cross-reacting between RSV subtypes A and B[14]. In the mRNA-1345 phase 3 efficacy trial, 36,557 participants aged ≥60 years from 22 countries were randomly assigned (1:1 ratio) to receive a single injection of mRNA-1345 50 μg or placebo. Efficacy of mRNA-1345 in the primary analysis (median follow-up: 3.7 months; range: 0.5–12.6 months) against the primary endpoints of RSV-associated lower respiratory tract disease with ≥2 or ≥3 signs or symptoms (RSV-LRTD-2+ and RSV-LRTD-3+) between 14 days and 12 months postinjection were 83.7% (95.88% confidence interval [CI],

[1]Moderna Inc, Cambridge, MA, USA. [2]Hospital Militar Central Cirujano Mayor Dr. Cosme Argerich, Buenos Aires, Argentina. [3]Spotlight Research Center LLC, Miami, FL, USA. [4]Translational and Clinical Research Institute, Newcastle University, Newcastle upon Tyne, UK. [5]NIHR Newcastle Clinical Research Facility, The Newcastle upon Tyne Hospitals NHS Foundation Trust, Newcastle upon Tyne, UK. ✉e-mail: lingyi.zheng@modernatx.com

66.0%–92.2%) and 82.4% (96.36% CI, 34.8%–95.3%); efficacy against the key secondary endpoint of RSV-associated acute respiratory disease (RSV-ARD) between 14 days and 12 months postinjection was 68.4% (95% CI, 50.9%–79.7%)[15]. An additional VE analysis was conducted when the global study cohort had ≥6 months of safety follow-up. Efficacy in the additional analysis (median follow-up: 8.6 months; range: 0.5–17.7 months) against RSV-LRTD-2+, RSV-LRTD-3+, and RSV-ARD were 63.3% (95% CI, 48.7%–73.7%), 63.0% (95% CI, 37.3%–78.2%), and 53.9% (95% CI, 40.5%–64.3%), respectively[16] (Fig. 1). These results confirmed persistent efficacy of a single dose of mRNA-1345 over a median 8.6 months follow-up in adults aged ≥60 years. Additionally, protection was generally consistent across RSV-A and RSV-B subtypes, age groups, frailty status, and in participants with pre-existing comorbidities[16] (Fig. S1).

The objective of this analysis was to evaluate nAb against RSV-A and -B subtypes, as well as IgG bAb to RSV preF or postfusion (postF), as (1) measured on Day 29 (hereafter, "Day 29 antibody marker"); (2) measured on Day 1 (day of receiving injection; hereafter, "baseline antibody marker"); (3) fold-rise from baseline antibody to Day 29 antibody marker (hereafter, "fold-rise antibody marker"), as correlates of risk (CoRs) and CoPs against primary and key secondary VE endpoints (hereafter, "RSV endpoints"). We prioritized the correlational analysis of the four antibody markers measured on Day 29. First, we applied a univariable Cox proportional hazards regression model to vaccine and placebo recipients to evaluate Day 29 antibody markers conditional on baseline risk factors as CoRs and CoPs against each RSV endpoint, respectively. Mediation analysis was conducted to evaluate how much VE was mediated by individual Day 29 antibody markers using the univariable Cox regression model[17,18]. We further evaluated individual Day 29 antibody markers as CoRs against each RSV endpoint caused by RSV-A and -B subtypes using a univariable Cox regression model to determine if consistent protections exist against RSV subtypes by Day 29 antibody markers. Finally, we applied similar methods to evaluate baseline antibody markers and fold-rise antibody markers as CoRs with each RSV endpoint using univariable Cox regression models.

## Results

### Immunogenicity subcohort, case-cohort set, and RSV endpoints

An immune correlate analysis was conducted based on participants in the case-cohort set (Fig. S2) who had RSV nAb and IgG bAb data assessed at Day 29 during the window (15–43) postinjection (vaccine or placebo), and had the RSV endpoint onset or censored more than 7 days after Day 29 (accounting for potential alteration of Day 29 antibody marker by likely natural RSV infection on the occurrence of RSV endpoints). The case-cohort set was comprised of a stratified random subcohort of participants (immunogenicity subcohort), plus all postinjection RSV-ARD cases (including early cases before 14 days postinjection)[19], and participants for the correlates analyses were denoted as the Day 29 case-cohort set (Fig. S3). Information about the case-cohort sampling design is provided in the supplement.

RSV endpoints (RSV-LRTD-2+, RSV-LRTD-3+, and RSV-ARD) were analyzed in the correlates analyses by the data cutoff (April 30, 2023), the same RSV endpoints studied in the additional analysis[16]. In the correlates analyses, per each RSV endpoint, cases were defined as participants in the Day 29 case-cohort set, with corresponding RSV endpoint onset more than 7 days after Day 29; non-cases were defined as participants in Day 29 case-cohort set with no evidence of the corresponding RSV endpoint onset up to the data cutoff. Overall, the correlates analyses included 2059 participants for analyzing RSV-LRTD-2+ (44 breakthrough cases in vaccine vs. 114 cases in placebo); 2071 participants for analyzing RSV-LRTD-3+ (19 breakthrough cases in vaccine vs. 49 cases in placebo); and 2045 participants for analyzing RSV-ARD (79 breakthrough cases in vaccine vs. 160 cases in placebo)

(Tables S1 and S2), respectively. The maximum event time of RSV endpoints following Day 29 was 345 days (the study period), which was used to estimate cumulative incidence and VE against RSV endpoints following Day 29.

### Participant demographics

Participant characteristics in the per-protocol immunogenicity subcohort (vaccine, $n = 1489$; placebo, $n = 327$) are displayed in Table S3, which consisted of all participants in the immunogenicity subcohort who had the RSV-ARD endpoint onset or censored more than 7 days after Day 29. Of 1816 participants in the subcohort, 45% were females, 45.5% were aged ≥75 years, 39.7% had a LRTD risk factor (COPD or chronic heart failure [CHF]) present at baseline (Day 1), 57.4% had ≥1 pre-existing comorbidities of interest (COPD, asthma, chronic respiratory disease, diabetes, CHF, advanced liver disease or renal disease), and 13.4% had a history of COVID-19. Overall, 56.5% were from the Northern hemisphere, 56.2% lived in high-income countries, and 45.9% were Hispanic or Latino. Participants in the per-protocol immunogenicity subcohort were well balanced in the vaccine and placebo groups according to these key baseline characteristics, but were not fully representative of the study cohort for the immune correlate analysis. Inverse probability of sampling weight (IPS-weight) was calculated based on the sampling design and applied to adjust the Day 29 case-cohort set in the correlates analyses (see Supplementary Information).

### Both Day 29 and baseline antibody levels were lower in RSV breakthrough cases versus non-cases

For all Day 29 and baseline antibody markers, 100% of vaccine and placebo recipients had antibody levels above the assay detection limits (Table S4, Fig. 2, Figs. S4–S6). Furthermore, the ratio of geometric mean (GM) values was approximately 0.5 to 0.8 comparing RSV-LRTD-2+ cases to non-cases for all Day 29 and baseline antibody markers in both vaccine and placebo groups (Table 1, Table S5). In the vaccine group, Day 29 RSV-A and RSV-B nAb geometric mean titers (GMTs) in RSV-LRTD-2+ cases were lower than those observed among non-cases; a similar trend was observed for Day 29 preF IgG and postF IgG geometric mean concentrations (GMCs) (Table 1, Table S5).

For Day 29 RSV-A nAb and preF IgG markers, the GM value was approximately 8–11 times higher for vaccine than placebo recipients by RSV-LRTD-2+ case status; for Day 29 RSV-B nAb and postF IgG markers, the respective GM value was about 6 times higher for vaccine than placebo recipients. Each pair of the 4 Day 29 antibody markers was highly correlated (e.g., preF bAb and RSV-A nAb, Spearman correlation $r = 0.88$; preF IgG and RSV-B nAb, $r = 0.79$; and RSV-A nAb and RSV-B nAb, $r = 0.76$; Figs. S7–S9), while baseline antibody markers were moderately correlated with each other (Figs. S10–S12). In addition, concordance analysis showed high concordance rates between RSV nAb and preF IgG markers by pooling baseline and Day 29 markers together (Fig. S13). For all Day 29 and baseline antibody markers, the GM value and ratio of GM (case vs. non-case; vaccine vs. placebo) by RSV-LRTD-3+ and RSV-ARD case status are shown in Tables S6–S7. Reverse cumulative distribution function curves for each Day 29 antibody marker are displayed in Figs. S14–S16.

### Day 29 antibody marker level inversely correlates with the risk of RSV endpoints

All Day 29 antibody markers were significantly inversely correlated with RSV-LRTD-2+ and RSV-ARD risk, and were consistently inversely correlated with RSV-LRTD-3+ risk (Table 2, Table S8). The covariate-adjusted hazard ratios (HR) of RSV-LRTD-2+ and RSV-ARD per each 10-fold increase in marker levels were similar (0.40–0.55). Given the small number of RSV-LRTD-3+ breakthrough cases ($n = 19$) in vaccine recipients, only RSV-A nAb showed significant inverse correlation with RSV-LRTD-3+ risk; the other three markers showed consistent inverse

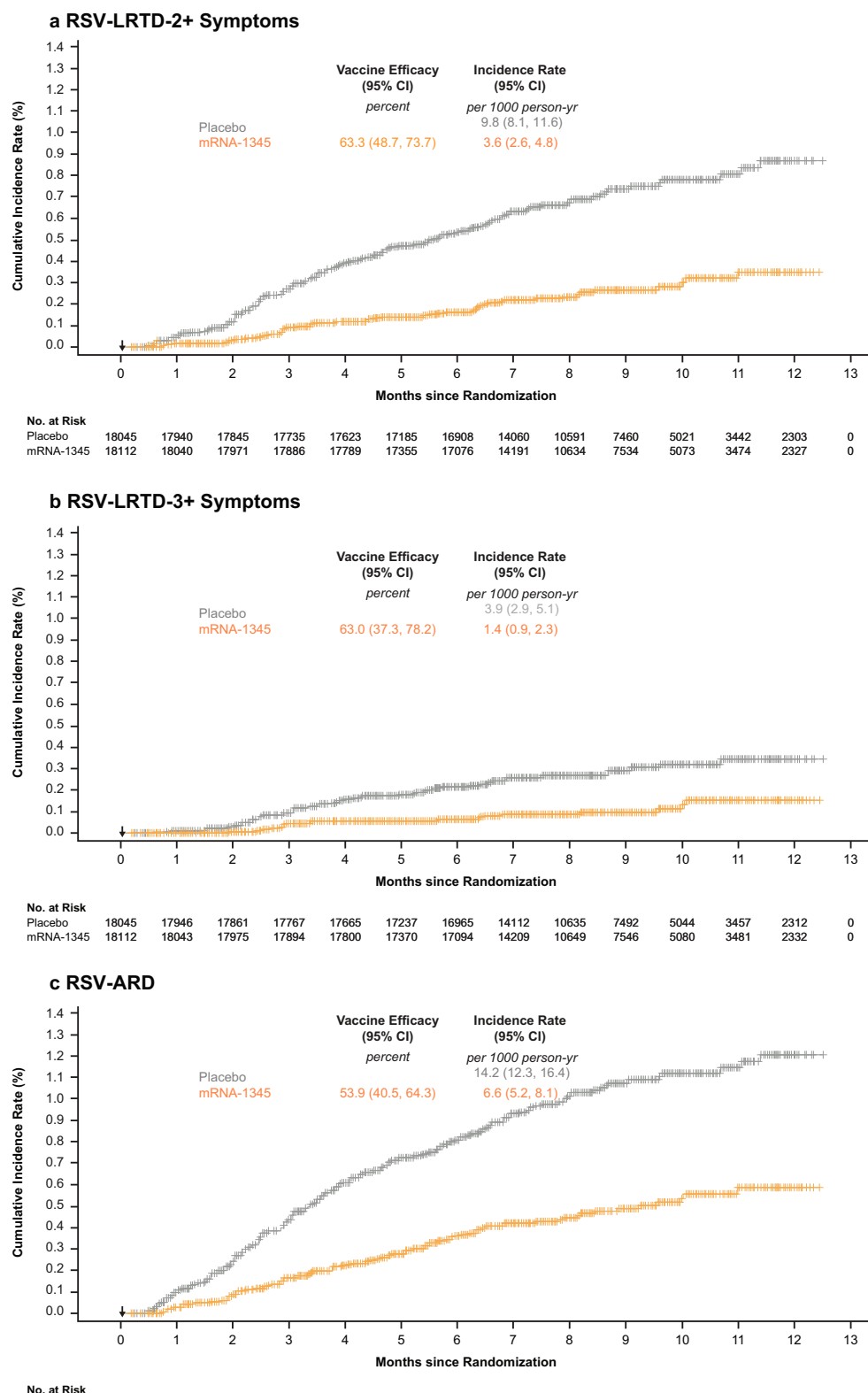

**Fig. 1 | Analysis of efficacy through 8.6 months of follow-up: cumulative incidence of RSV-LRTD and RSV-ARD in PPE set.** Shown are the **a** cumulative incidence of RSV-LRTD-2+ symptoms, **b** cumulative incidence of RSV-LRTD-3+ symptoms, and **c** cumulative incidence of RSV-ARD. Only the first episodes occurring between 14 days and 12 months postinjection were included in the analysis (PPE set). All participants who had been randomly assigned, received the vaccine or placebo, completed ≥1 visit or surveillance contact 14 days after injection, and had no major protocol deviation that would affect the efficacy outcomes were included. In each panel, the arrow indicates when the injection was administered (Day 1). The cumulative incidence is based on the Kaplan–Meier method, and the incidence rate was defined as the number of participants with a case, divided by the number of participants at risk, with adjustment for person-years. Tick marks indicate censored data. ARD acute respiratory disease, LRTD lower respiratory tract disease, PPE per-protocol efficacy, RSV respiratory syncytial virus.

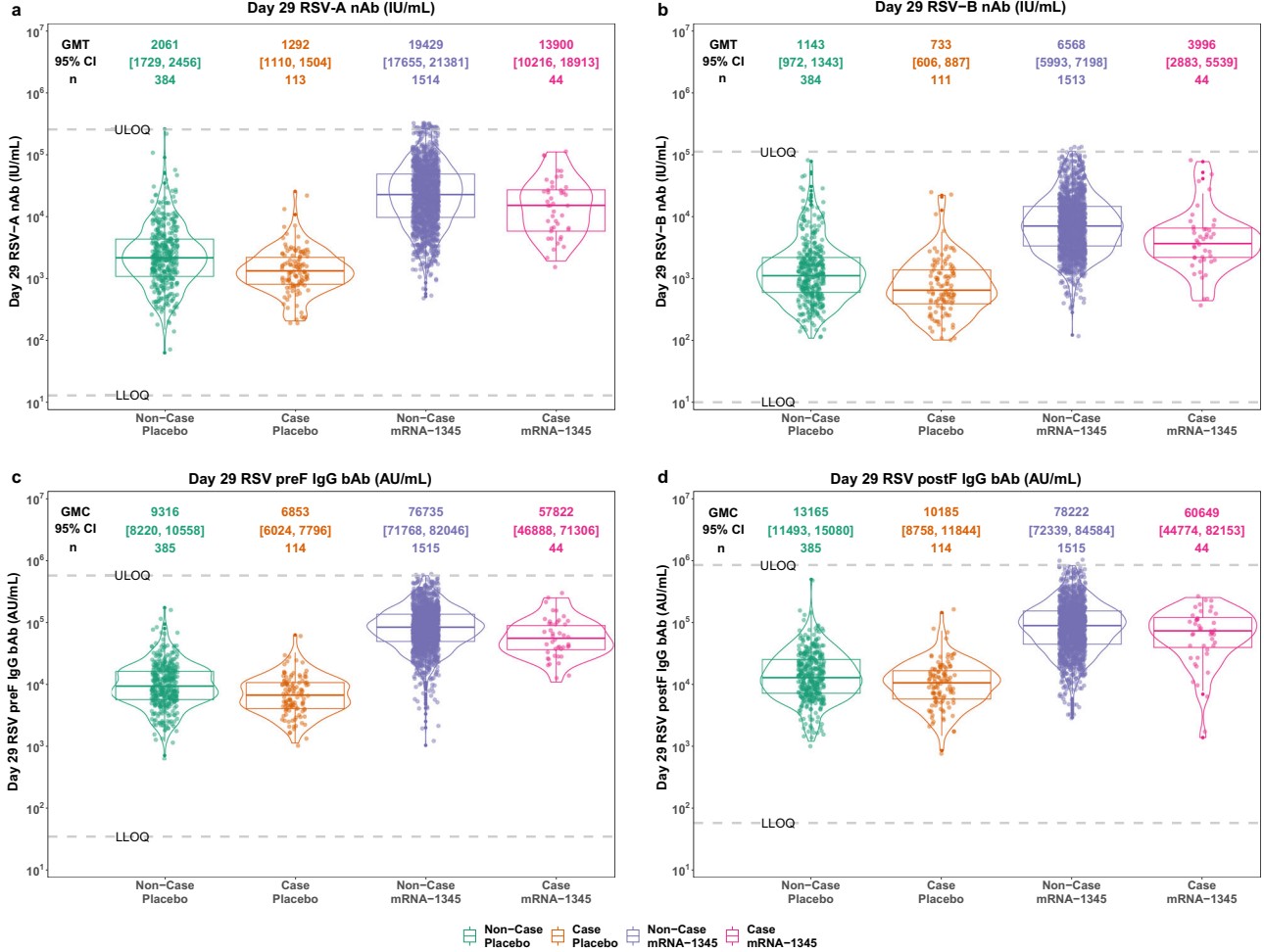

**Fig. 2 | Day 29 RSV nAb (IU/mL) and IgG bAb (AU/mL) by RSV-LRTD-2+ case status and by vaccine and placebo in the Day 29 case-cohort set. a** RSV-A nAb. **b** RSV-B nAb. **c** RSV preF IgG bAb. **d** RSV postF IgG bAb. The violin-box plot is composed of an interior box plot and rotated probability density plots (estimated by a default Gaussian kernel density estimator) of the antibody marker data on each side. In the box plot, the middle line and the lower and upper horizontal edges represent the 50th, 25th, and 75th percentile of antibody titers or concentrations, and the vertical whiskers represent the distance from the 25th (or 75th) percentiles of antibody titers or concentrations and the minimum (or maximum) antibody titers or concentrations within the 25th (or 75th) percentile of antibody level minus (or plus) 1.5 times the interquartile range. The GMT or GMC level and the corresponding 95% confidence interval are adjusted by the IPS-weight. ARD acute respiratory disease, bAb binding antibody, GMC geometric mean concentration, GMT geometric mean titer, IgG immunoglobulin G, IPS inverse probability of sampling, LLOQ lower limit of quantification, LRTD lower respiratory tract disease, nAb neutralizing antibody, postF postfusion, preF prefusion, RSV respiratory syncytial virus, ULOQ upper limit of quantification.

correlations, albeit this did not reach statistical significance ($p$-values > 0.05). Table S9 shows that the correlation of each Day 29 antibody marker and each RSV endpoint was not statistically different between vaccine and placebo.

The estimated cumulative incidence of each RSV endpoint by placebo group and vaccine tertile subgroups (defined by Day 29 antibody marker tertiles in vaccine recipients) indicated that RSV risk decreased from placebo through low, medium, and high vaccine tertile subgroups for each marker (except postF IgG) (Figs. S17–S19). The estimated instantaneous hazard rate of each RSV endpoint by placebo and vaccine tertile subgroups for each Day 29 antibody marker is shown in Figs. S20–S22, depicting coherent trends to support RSV risk inversely correlating with increased vaccine tertiles. Covariate-adjusted HR of RSV-LRTD-2+ for vaccine tertile subgroups versus placebo showed decreased risk with increasing vaccine tertile subgroups for RSV-A nAb, RSV-B nAb, and preF IgG, but not for postF IgG (Fig. 3). For RSV-LRTD-2+, $p$-values and family-wise error rate (FWER)-adjusted $p$-values for all vaccine tertiles by each Day 29 antibody markers were significant ($p < 0.05$). Covariate-adjusted HRs of RSV-LRTD-3+ and RSV-ARD in vaccine tertile subgroups are shown in

Table S10, demonstrating that RSV risk decreased with increments of RSV nAb and preF IgG levels.

## Day 29 antibody marker level positively correlates with VE for RSV endpoints

Fig. S23 shows that the marginal treatment effect for each RSV endpoint was consistent with clinical VE, and that the treatment effect conditional on each Day 29 bAb and nAb marker (except postF IgG) was not significant and was nearly mediated (i.e., estimated HR close to 1; although to a lesser extent for RSV-LRTD-3+ due to a smaller number of break-through cases [$n = 19$] in the vaccine group), indicating the risk of getting each RSV endpoint can be predicted by Day 29 antibody marker independently of treatment (vaccine or placebo). By the Prentice surrogate endpoint criteria[20], all RSV nAb and preF IgG markers measured at Day 29 were supported as surrogate endpoints (i.e., CoPs) for all RSV endpoints; RSV-A nAb and preF IgG showed the strongest evidence (small HRs of RSV endpoints per 10-fold increase marker levels and large $p$-values for nearly mediated conditional treatment effect).

Figure 4 displays further correlates analysis specifically for RSV-LRTD-2+ and shows the estimated (1) cumulative incidence of RSV-

**Table 1 | Baseline and Day 29 antibody marker GMT or GMC level by case/non-case strata and by vaccine and placebo in the Day 29 case-cohort set for RSV-LRTD-2+**

| Immunologic marker | Treatment group or ratio of GM | Cases GMC or GMT (95% CI) | | | Non-cases GMC or GMT (95% CI) | | | Ratio of GM (cases/non-cases) (95% CI) | |
|---|---|---|---|---|---|---|---|---|---|
| | | *n* | Baseline | Day 29 | *n* | Baseline | Day 29 | Baseline | Day 29 |
| RSV-A nAb (IU/mL) | mRNA-1345 | 44 | 1014 (757, 1358) | 13,900 (10,216, 18,913) | 1516 | 2231 (2040, 2441) | 19,429 (17,655, 21,381) | 0.5 (0.3, 0.6) | 0.7 (0.5, 1.0) |
| | Placebo | 114 | 1303 (1117, 1520) | 1292 (1110, 1504) | 385 | 1950 (1636, 2325) | 2061 (1729, 2456) | 0.7 (0.5, 0.8) | 0.6 (0.5, 0.8) |
| | Ratio of GM (mRNA-1345/Placebo) (95% CI) | | 0.8 (0.6, 1.1) | 10.8 (7.6, 15.2) | | 1.1 (0.9, 1.4) | 9.4 (7.7, 11.5) | | |
| RSV-B nAb (IU/mL) | mRNA-1345 | 44 | 627 (474, 830) | 3996 (2883, 5539) | 1516 | 1250 (1143, 1367) | 6568 (5993, 7198) | 0.5 (0.4, 0.7) | 0.6 (0.4, 0.9) |
| | Placebo | 114 | 716 (596, 860) | 733 (606, 887) | 385 | 1091 (941, 1266) | 1143 (972, 1343) | 0.7 (0.5, 0.8) | 0.6 (0.5, 0.8) |
| | Ratio of GM (mRNA-1345/Placebo) (95% CI) | | 0.9 (0.6, 1.2) | 5.5 (3.7, 8.0) | | 1.1 (1.0, 1.4) | 5.7 (4.8, 6.9) | | |
| RSV preF IgG bAb (AU/mL) | mRNA-1345 | 44 | 5827 (4613, 7360) | 57,822 (46,888, 71,306) | 1516 | 9509 (8932, 10,123) | 76,735 (71,768, 82,046) | 0.6 (0.5, 0.8) | 0.8 (0.6, 0.9) |
| | Placebo | 114 | 7041 (6153, 8057) | 6853 (6024, 7796) | 385 | 8747 (7776, 9839) | 9316 (8220, 10,558) | 0.8 (0.7, 1.0) | 0.7 (0.6, 0.9) |
| | Ratio of GM (mRNA-1345/Placebo) (95% CI) | | 0.8 (0.6, 1.1) | 8.4 (6.6, 10.8) | | 1.1 (1.0, 1.2) | 8.2 (7.1, 9.5) | | |
| RSV postF IgG bAb (AU/mL) | mRNA-1345 | 44 | 8652 (6796, 11,015) | 60,649 (44,774, 82,153) | 1516 | 12,885 (11,988, 13,848) | 78,222 (72,339, 84,584) | 0.7 (0.5, 0.9) | 0.8 (0.6, 1.1) |
| | Placebo | 114 | 10,326 (8887, 11,998) | 10,185 (8758, 11,844) | 385 | 12,405 (10,859, 14,170) | 13,165 (11,493, 15,080) | 0.8 (0.7, 1.0) | 0.8 (0.6, 0.9) |
| | Ratio of GM (mRNA-1345/Placebo) (95% CI) | | 0.8 (0.6, 1.1) | 6.0 (4.2, 8.4) | | 1.0 (0.9, 1.2) | 5.9 (5.1, 6.9) | | |

*bAb* binding antibody, *CI* confidence interval, *GM* geometric mean, *GMC* geometric mean concentration, *GMT* geometric mean titer, *IgG* immunoglobulin G, *LRTD* lower respiratory tract disease, *nAb* neutralizing antibody, *postF* postfusion, *preF* prefusion, *RSV* respiratory syncytial virus.

LRTD-2+ during the study period across a range of assigned marker levels by vaccine and placebo; (2) controlled VE against RSV-LRTD-2+ during the study period across a range of assigned marker levels; and (3) cumulative incidence of RSV-LRTD-2+ during the study period above a range of assigned marker levels (thresholds) by Day 29 RSV-A nAb and RSV preF IgG markers of vaccinees (see Supplementary Information), respectively. Specifically, for both Day 29 RSV-A nAb and preF IgG, the estimated cumulative incidence of RSV-LRTD-2+ by vaccine and placebo group was similar and overlapped by bootstrap pointwise 95% CIs; estimated VE increased as the antibody marker level increased. Additionally, we conducted a threshold analysis for each Day 29 antibody marker by analyzing subgroups with antibody levels greater than or equal to a certain value, supporting that the higher postvaccination immune response of subgroups correlated with decreased risk of RSV disease. Further CoR and CoP analyses for each RSV endpoint by Day 29 antibody marker are shown in Figs. S24–S29.

**Day 29 antibody markers mediate the majority of mRNA-1345 VE against RSV endpoints**
For all Day 29 antibody markers, the majority of marker levels (>90%) had overlapping distributions for vaccine and placebo recipients;

therefore, it was feasible to assess how much VE for each RSV endpoint could be mediated by these markers using an adapted approach from Benkeser et al.[21] (see Supplementary Information). The estimated proportion of VE mediated through the antibody markers at Day 29 was highest for RSV-A nAb and preF IgG for all RSV endpoints; Day 29 RSV-B nAb showed comparable or moderate VE mediation against each RSV endpoint compared to Day 29 RSV-A nAb; postF IgG showed limited VE mediation against each RSV endpoint compared to the other 3 Day 29 markers (Fig. S30).

**CoR analysis of Day 29 antibody markers against RSV endpoints by RSV subtypes A and B**
We further studied Day 29 antibody markers as CoRs against each RSV endpoint by RSV-A and -B subtypes (Fig. S31). The results of the covariate-adjusted HR of each RSV endpoint by RSV subtype per 10-fold increase in Day 29 preF IgG demonstrated that this antibody marker was very consistent as a CoR for all RSV endpoints regardless of subtype (Table S11). Day 29 postF IgG showed less consistent evidence as CoRs compared with preF IgG. Overall, RSV-A and RSV-B nAb were the best CoRs for RSV subtype-matched endpoints, but RSV-A nAb more consistently showed inverse correlates with RSV endpoints by both subtypes versus RSV-B nAb (Table S11).

**Table 2 | Covariate-adjusted hazard ratios of each RSV endpoint per 10-fold increase in each Day 29 antibody marker in vaccine and placebo recipients in the Day 29 case-cohort set**

| Endpoint | Immunologic marker | No. cases/ No. at-risk[*] | Attack rate | Hazard ratio per 10-fold increase point Est. (95% CI) | P-value (2-sided) | FWER adjusted P-value[†] |
|---|---|---|---|---|---|---|
| RSV-LRTD-2+ | RSV-A nAb (IU/mL) | 44/17,555 | 0.0025 | 0.44 (0.30,0.65) | <0.001 | <0.001 |
| | RSV-B nAb (IU/mL) | 44/17,553 | 0.0025 | 0.42 (0.25,0.72) | 0.001 | 0.002 |
| | RSV preF IgG bAb (AU/mL) | 44/17,600 | 0.0025 | 0.40 (0.23,0.68) | 0.001 | 0.002 |
| | RSV postF IgG bAb (AU/mL) | 45/17,600 | 0.0026 | 0.55 (0.33,0.89) | 0.016 | 0.016 |
| | Placebo | 115/17,470 | 0.0066 | | | |
| RSV-LRTD-3+ | RSV-A nAb (IU/mL) | 19/17,594 | 0.0011 | 0.41 (0.20,0.84) | 0.014 | 0.056 |
| | RSV-B nAb (IU/mL) | 19/17,593 | 0.0011 | 0.55 (0.25,1.24) | 0.151 | 0.183 |
| | RSV preF IgG bAb (AU/mL) | 19/17,637 | 0.0011 | 0.47 (0.21,1.04) | 0.063 | 0.183 |
| | RSV postF IgG bAb (AU/mL) | 19/17,637 | 0.0011 | 0.60 (0.28,1.28) | 0.183 | 0.183 |
| | Placebo | 51/17,395 | 0.0029 | | | |
| RSV-ARD | RSV-A nAb (IU/mL) | 79/17,601 | 0.0045 | 0.45 (0.28,0.71) | 0.001 | 0.003 |
| | RSV-B nAb (IU/mL) | 80/17,546 | 0.0046 | 0.42 (0.25,0.73) | 0.002 | 0.003 |
| | RSV preF IgG bAb (AU/mL) | 80/17,600 | 0.0045 | 0.36 (0.21,0.63) | <0.001 | <0.001 |
| | RSV postF IgG bAb (AU/mL) | 80/17,600 | 0.0045 | 0.46 (0.28,0.77) | 0.003 | 0.003 |
| | Placebo | 162/17,463 | 0.0093 | | | |

Age, LRTD at-risk, and baseline risk score are adjusted in the Cox PH model. The maximum failure event time post Day 29 visit is 345 days.

*ARD* acute respiratory disease, *bAb* binding antibody, *FWER* family-wise error rate, *IgG* immunoglobulin G, *IPS* inverse probability of sampling, *LRTD* lower respiratory tract disease, *nAb* neutralizing antibody, *PH* proportional hazard, *postF* postfusion, *preF* prefusion, *RSV* respiratory syncytial virus.

*No. cases: Estimated number of participants who received vaccine or placebo with RSV endpoint (RSV-LRTD-2+, RSV-LRTD-3+, RSV-ARD) onset during the study period. No. at-risk: Estimated number of participants who received vaccine or placebo not experiencing RSV endpoint (RSV-LRTD-2+, RSV-LRTD-3+, and RSV-ARD) onset by 7 days after the Day 29 visit. The estimated no. cases and no. at-risk differs slightly across endpoints due to the variability of the number of participants with observed eligible antibody marker data.

†FWER (family-wise error rate)-adjusted *p*-values were calculated by each RSV endpoint using the Hommel method.

Baseline risk factors are adjusted in the univariable IPS-weighted Cox PH regression model, including the actual stratification factors age and LRTD at-risk and baseline risk score.

## CoR analysis of baseline antibody markers against RSV endpoints

Using the same Cox regression model as above, we also investigated whether baseline antibody markers were CoRs against each RSV endpoint. All baseline antibody markers except postF IgG were significantly inversely correlated with all RSV endpoints (Fig. S32). Importantly, the treatment effect conditional on any individual baseline antibody markers was similar to the marginal treatment effect (estimated HR in vaccine: 0.36 [0.24−0.55] for RSV-LRTD-2+, 0.27 [0.12−0.61] for RSV-LRTD-3+, and 0.52 [0.36−0.76] for RSV-ARD). This result shows that the probability of RSV depends on both baseline antibody markers and treatment, suggesting that pre-vaccination (baseline) antibody markers alone are not useful for predicting VE.

## CoR analysis of fold-rise antibody markers against RSV endpoints

Lastly, we studied whether fold-rise antibody markers are CoRs and mediate protection against each RSV endpoint. As opposed to the above baseline and Day 29 antibody marker analyses, only vaccine recipients in the Day 29 case-cohort set were used to study the correlates of fold-rise antibody markers, since the fold-rise in any antibody markers measured at Day 29 compared to Day 1 (baseline) was nearly unchanged (close to 1) in placebo recipients (Tables S12−S14, Figs. S33−S35). For each antibody marker, fold-rise in vaccine recipients decreased as the baseline marker level increased; fold-rise of RSV-A nAb was significantly higher in breakthrough cases versus non-cases in vaccinees for each RSV endpoint. Fold-rise in other antibody markers was not significantly or consistently different in breakthrough versus non-cases in vaccinees for any RSV endpoint. Moreover, Cox-based model results showed that only fold-rise in RSV-A nAb was significant; the other fold-rise markers showed consistent positive correlates with RSV endpoints to a different extent than RSV-A nAb (Fig. S36). Both GM and model results provide consistent evidence to support fold-rise antibody markers positively correlating with the risk of RSV endpoints.

A potential explanation of this paradoxical finding is that, due to the negative correlation between fold-rise and baseline marker level, the positive correlation of fold-rise in antibody markers with the risk of RSV endpoints could be largely masked by the inverse correlation of baseline level with the risk of RSV endpoints.

## Discussion

Our comprehensive immune correlates analysis of the phase 3 efficacy trial of mRNA-1345 versus placebo demonstrated that RSV nAb (A and B subtypes) and RSV preF IgG bAb markers assessed at Day 29 were significantly inversely correlated with primary and key secondary endpoints (RSV-LRTD-2+ and RSV-ARD) and were also strongly and consistently inversely correlated with another primary endpoint (RSV-LRTD-3+). The treatment effect for each RSV endpoint, conditional on Day 29 RSV-A nAb, RSV-B nAb, and preF IgG markers, was not significant and was nearly mediated by these markers (estimated HR close to 1). The interaction effect between treatment and these antibody markers was also not significant, supporting that RSV risk only depends on antibody level, regardless of whether the antibody is elicited by vaccine or natural RSV infection. Therefore, by the Prentice criteria, RSV nAb and preF IgG markers are supported as surrogate endpoints/CoPs and can be considered as surrogate markers for RSV endpoints. Conversely, while Day 29 postF IgG levels also had strong inverse correlations with RSV endpoints, the treatment effect against RSV-LRTD-2+, conditional on postF IgG levels, was somewhat closer to the marginal treatment effect. Importantly, evidence of being a CoR for Day 29 postF IgG under quantitative and qualitative univariable Cox regression models was not consistent, indicating that postF IgG is not a CoP. The complementary analysis of postF IgG echoed the historical lessons of the failure of postF-based RSV vaccines and further supports the development strategy of mRNA-1345, which successfully protects against RSV disease by targeting RSV prefusion F protein[22,23].

The estimated cumulative incidence of each RSV endpoint across a range of Day 29 RSV-A nAb titers (or preF IgG levels) decreased

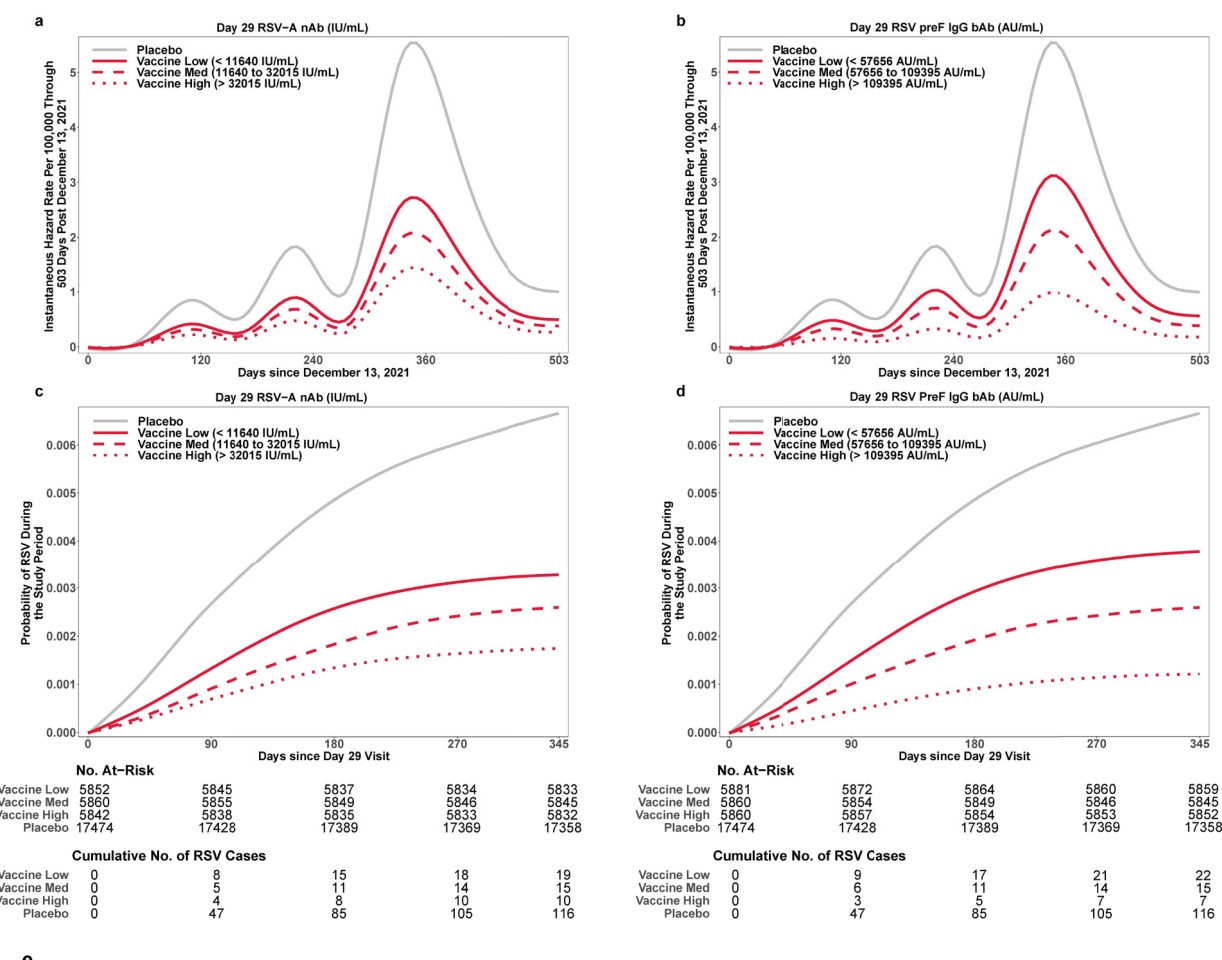

| Immunologic Marker | Tertile | No. Cases/ No. At-Risk* | Attack Rate | Hazard Ratio Point Est. (95% CI) | P-Value (2-sided) | FWER Adjusted P-Value† |
|---|---|---|---|---|---|---|
| RSV-A nAb (IU/mL) | Low (< 11640 AU/mL) | 19/5852 | 0.0032 | 0.44 (0.25,0.77) | 0.004 | 0.012 |
| | Med (11640 to 32015 AU/mL) | 15/5860 | 0.0026 | 0.37 (0.20,0.69) | 0.002 | 0.008 |
| | High (> 32015 AU/mL) | 10/5842 | 0.0017 | 0.27 (0.13,0.54) | <0.001 | <0.001 |
| RSV-B nAb (IU/mL) | Low (< 3833 AU/mL) | 23/5849 | 0.0039 | 0.49 (0.29,0.86) | 0.012 | 0.012 |
| | Med (3833 to 10371 AU/mL) | 15/5858 | 0.0026 | 0.37 (0.20,0.68) | 0.001 | 0.006 |
| | High (> 10371 AU/mL) | 6/5846 | 0.0010 | 0.17 (0.07,0.41) | <0.001 | <0.001 |
| RSV preF IgG bAb (AU/mL) | Low (< 57656 AU/mL) | 22/5881 | 0.0037 | 0.48 (0.28,0.83) | 0.008 | 0.012 |
| | Med (57656 to 109395 AU/mL) | 15/5860 | 0.0026 | 0.37 (0.20,0.69) | 0.002 | 0.008 |
| | High (> 109395 AU/mL) | 7/5860 | 0.0012 | 0.20 (0.09,0.45) | <0.001 | <0.001 |
| RSV postF IgG bAb (AU/mL) | Low (< 57339 AU/mL) | 16/5870 | 0.0027 | 0.35 (0.19,0.65) | 0.001 | 0.006 |
| | Med (57339 to 120009 AU/mL) | 15/5870 | 0.0026 | 0.40 (0.22,0.73) | 0.003 | 0.009 |
| | High (> 120009 AU/mL) | 13/5861 | 0.0022 | 0.34 (0.18,0.64) | 0.001 | 0.006 |
| Placebo | | 116/17474 | 0.0066 | | | |

Age, LRTD at-risk, and baseline risk scores were adjusted in the Cox PH model. The maximum failure event time after the Day 29 visit is 345 days.

*No. of cases: Estimated number of participants receiving vaccine or placebo with an RSV-LRTD-2+ endpoint onset during the study period. No. at-risk: Estimated number of participants receiving vaccine or placebo not experiencing an RSV-LRTD-2+ endpoint onset by 7 days after the Day 29 visit. The estimated no. of cases and no. at-risk are slightly different due to the variability of the number of participants with observed eligible antibody marker data.

†FWER-adjusted P-values are calculated by each RSV endpoint using the Hommel method.

**Fig. 3 | RSV-LRTD-2+ risk by placebo and antibody marker level in vaccine recipients in the Day 29 case-cohort set.** Covariate-adjusted instantaneous hazard rate and cumulative incidence of RSV-LRTD-2+ by placebo and by low, medium, and high tertile of Day 29 RSV nAb or IgG bAb marker level in vaccine recipients. (**a**) and (**c**) for Day 29 RSV-A nAb. (**b**) and (**d**) for Day 29 RSV preF IgG bAb. **e** Day 29 RSV-A and RSV-B nAb and RSV preF and postF IgG bAb. Baseline risk factors are adjusted in the univariable (qualitative) IPS-weighted Cox PH regression model, including the actual stratification factors age and LRTD at-risk and baseline risk score. bAb binding antibody, FWER family-wise error rate, IgG immunoglobulin G, IPS inverse probability of sampling, LRTD lower respiratory tract disease, nAb neutralizing antibody, PH proportional hazard, point est. point estimate, postF postfusion, preF prefusion, RSV respiratory syncytial virus.

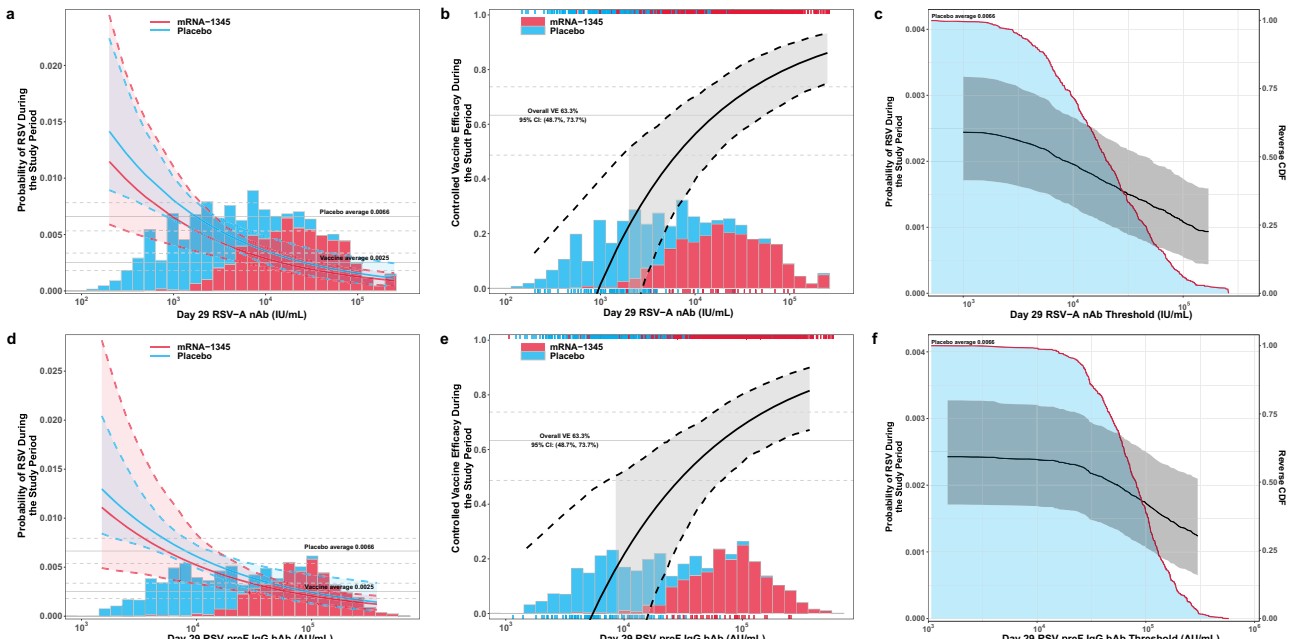

**Fig. 4 | Further CoR/CoP analysis for RSV-LRTD-2+ by Day 29 RSV-A nAb and Day 29 RSV preF IgG bAb, respectively. a** and **d** Solid red and blue curves demonstrate point estimates of the covariate-adjusted cumulative incidence of RSV-LRTD-2+ during the study period for vaccine and placebo recipients across a range of assigned antibody titers or concentration levels (within 0.5th–99.5th percentiles of observed antibody values in vaccine and placebo groups). Dashed red and blue curves, along with the shades, represent the bootstrap pointwise 95% CIs. Solid and dashed horizontal gray lines represent the point estimates and 95% CIs of the average covariate-adjusted cumulative incidence of RSV-LRTD-2+ in vaccine and placebo recipients. **b** and **e** Solid black curve shows the point estimate of controlled VE across a range of assigned antibody titers or concentration levels (within 0.5th–99.5th percentiles of observed antibody values in vaccine and placebo groups), dashed black curves demonstrate the bootstrap pointwise 95% CIs. Rug lines below and above represent RSV-LRTD-2+ cases and non-cases by vaccination status, respectively. The shaded gray area between dashed curves highlights the VE above the median of the antibody level in placebo recipients. Solid and dashed horizontal gray lines are point estimates and 95% CIs of clinical VE in the additional analysis. **c** and **f** Red curve along the blue area represents the reverse cumulative density function values for the observed antibody marker values in vaccine recipients. The black curve is the covariate-adjusted cumulative incidence of RSV-LRTD-2+ during the study period above a range of antibody marker levels (below the 97.5th percentile of observed antibody values in the vaccine group). The shadowed gray area is the bootstrap pointwise 95% CIs. The stacked histogram of the observed antibody marker titers or concentration levels by vaccination status overlayed on the bottom of cumulative incidence plots (**a**, **d**) and VE plots (**b**, **e**). Baseline covariates of age, LRTD at-Risk, and baseline risk score are adjusted in the IPS-weighted Cox regression model. bAb binding antibody, CDF cumulative distribution function, CI confidence interval, CoP correlate of protection, CoR correlate of risk, IgG immunoglobulin G, IPS inverse probability of sampling, LRTD lower respiratory tract disease, nAb neutralizing antibody, postF postfusion, preF prefusion, RSV respiratory syncytial virus, VE vaccine efficacy.

steadily as the antibody marker value increased in both vaccine and placebo groups (bootstrapped 95% CIs overlapped). A larger gap in the estimated cumulative incidence of RSV-LRTD-2+ and RSV-LRTD-3+ between vaccine and placebo groups was observed for Day 29 RSV-B nAb and postF IgG, consistent with the less mediated treatment effect conditional on the corresponding markers, though the 95% CIs still overlapped. For each RSV endpoint and Day 29 antibody marker, estimated VE[24,25] increased monotonically as the antibody marker level increased, and the 95% CI of VE was narrowest by RSV-A nAb (comparable to preF IgG), and narrower when comparing RSV-A nAb to RSV-B nAb and preF IgG to postF IgG. For all Day 29 antibody markers, the CI band of VE for RSV-LRTD-3+ was largest due to a relatively small number of vaccine breakthrough cases ($n = 19$). Moreover, instead of the nonparametric threshold method[26], we used a causal inference method to estimate the expected cumulative incidence of each RSV endpoint above a range of marker levels for each Day 29 antibody using the univariable Cox regression model. The advantage of our method is to guarantee the monotone decreasing trend of the cumulative risk with marker threshold (similar to the method in Van der Laan et al.[27]) and narrower CIs. However, it is notable that multiple cases in the vaccine group had high nAb titers or preF IgG levels (Figs. S4–S6), indicating that the nAb or preF IgG marker may not be a mechanistic CoP, and that this subgroup with high nAb titers or preF IgG levels cannot attain a very low level of RSV risk (Figs. S24–S29). Additionally, many non-cases in the vaccine group had relatively low nAb titer or

preF IgG levels, indicating that other types of antibody markers may exist to protect against RSV disease. Future studies can evaluate additional antibody responses generated by alternate vaccine approaches (e.g., live attenuated vaccines or vaccines containing more than one antigen) as potential CoPs for RSV disease.

Because all baseline marker levels were far above the assay lower detection limit (Table S4), and Day 29 antibody marker levels in placebo recipients were largely overlapped with those in vaccine recipients (>10%), the placebo group can to some extent be considered as another "vaccine" (natural infection treatment). Therefore, we proposed a new mediation analysis approach[28] adapted from the Benkeser et al. method,[21] pooling vaccine and placebo recipients in the analysis to quantify the indirect, direct VE, and VE mediator (mediated proportion of VE) by Day 29 antibody markers. Our mediation analysis showed that RSV-A nAb and preF IgG similarly mediated the majority or full VE for RSV endpoints, while a relatively lower proportion of VE was mediated by RSV-B nAbs versus RSV-A nAbs. Of all antibody markers, postF IgG had the lowest mediated VE against each RSV endpoint. Notably, the mediated proportion of VE was consistent with the conditional treatment effect on the Day 29 antibody marker. This causal inference analysis result showed that RSV nAb and preF IgG markers measured at Day 29 can be considered as CoPs, especially Day 29 RSV-A nAbs and preF IgG.

All above correlational analyses were against all RSV endpoints caused by any RSV subtypes (either A or B). A naturally interesting

question was to understand the extent each antibody marker could consistently protect against RSV subtypes. Therefore, we further evaluated Day 29 antibody markers as CoRs against the RSV endpoints caused by RSV-A and -B subtypes. This exploratory analysis showed that RSV-A and RSV-B nAbs had inverse correlates with RSV endpoints regardless of subtype; however, RSV-A nAbs were more consistently inversely correlated than RSV-B nAbs, supporting this antibody marker as potentially providing more consistent protection against RSV regardless of the exact subtype. Importantly, the analysis supported that preF IgG consistently demonstrated protection against RSV subtypes A or B, supporting this antibody marker as a common CoP for RSV endpoints based on subtype. One possible explanation for these findings is that mRNA-1345 was engineered based on an RSV-A2 strain protein sequence for stabilizing the preF conformation, which is preserved across strains and antigenic subtypes (RSV-A and RSV-B)[29–32] and may induce higher nAbs against RSV- than -B[33,34]; however, the differences in assays limit any direct comparisons between RSV subtypes.

We also investigated baseline and fold-rise antibody markers as CoRs against RSV endpoints. These analyses found that all baseline antibody markers had strong inverse correlation with RSV endpoints as Day 29 antibody markers; which coincided with the GMT (or GMC) ratio for baseline nAb titers (or preF IgG levels) for cases versus non-cases in the vaccine or placebo groups being lower than that observed for Day 29 antibody markers (Table S5–S7). Of note, the ratio of cases versus non-cases by an RSV endpoint (e.g., RSV-LRTD-3+) for some antibody markers may be cross-over one, which may result in large variability due to the relevant small number of cases. Nevertheless, baseline antibody markers were less reliable in predicting VE compared to Day 29 markers because the corresponding conditional treatment effect was largely unchanged compared to the marginal treatment effect. Conversely, all fold-rise antibody markers were positively correlated with RSV endpoints, and correlation strengths between fold-rise markers and RSV endpoints were comparable to Day 29 markers. However, such positive correlation between fold-rise antibody markers and RSV endpoints was very likely reversely masked by baseline antibody markers due to a high negative correlation between fold-rise and baseline antibody markers (Spearman correlation <−0.8). Fold-rise in antibody levels is commonly used to evaluate vaccine-elicited immune responses and is often considered a secondary endpoint in vaccine trials. However, in this study, the absolute antibody titer/level was more reasonably correlated with the protection from RSV disease as compared to the corresponding fold-rise antibody marker. As we only included vaccine recipients to study the fold-rise antibody markers as CoRs of RSV endpoints, we cannot implement the mediation analysis or apply the Prentice criterion to further evaluate the likelihood of fold-rise markers being CoPs. Additionally, it is difficult to meaningfully interpret such positive correlations between fold-rise antibody markers with RSV endpoints.

Our immune correlates analyses have many advantages in quantifying the strength of immune markers as CoPs. First, the mRNA-1345 phase 3 efficacy trial was a randomized, double-blinded, placebo-controlled study with a pre-specified immune correlate analysis plan, bolstering the credibility of the interpretation of statistical results (HRs, p-values, and CIs) for making conclusions. Secondly, these analysis methods were largely consistent with the correlational analysis conducted in the mRNA-1273 COVID-19 phase 3 efficacy study[2,7]; additionally, we applied the Prentice criteria to quantify the strength of immune antibody markers as CoPs. We proposed a parametric causal approach to estimate the cumulative incidence above a certain marker level, with narrower CIs and more meaningful interpretations than the nonparametric threshold analysis approach. Importantly, we also proposed a new mediation analysis method to better evaluate how much VE could be mediated by immune markers. Moreover, we investigated whether RSV nAb and IgG bAb could consistently protect against RSV subtypes (A or B), with the results suggesting that both markers could be used as a common CoP for RSV endpoints, while postF IgG should not. Lastly, we applied univariable Cox regression models to investigate baseline, Day 29, and fold-rise antibody markers as CoRs and CoPs against RSV endpoints synthetically, and established systematic concepts that vaccine-elicited antibody levels (e.g., Day 29 antibody markers) were CoPs for VE of RSV endpoints, while baseline antibody markers were CoRs but not reliably predictable of VE, and fold-rise antibody markers were positively correlated with RSV endpoints.

Limitations include that the current establishment of CoP by RSV nAb and IgG bAb markers might not be mechanistic CoPs[35] due to the inability to measure antibody levels at the illness day as well as the inability to observe RSV exposures and to condition on them in the analysis; future work is planned to assess RSV nAbs and bAbs over time to conduct an exposure-proximal correlate analysis. The number of RSV-LRTD-3+ breakthrough cases in vaccine recipients was relatively small ($n = 19$); therefore, the strength of statistical significance of nAb and IgG bAb as CoPs against RSV-LRTD-3+ is weaker than that for RSV-LRTD-2+ and RSV-ARD. Future work could further evaluate nAb and IgG bAb markers as CoPs of mRNA-1345 using a later data cutoff date. Finally, we studied co-primary and key secondary RSV endpoints with immune response markers independently, ignoring the correlations between these endpoints and potentially simplifying the correlate analysis; additionally, FWER-adjusted p-values were calculated across antibody markers by each RSV endpoint, which may limit the interpretation of immune response antibody markers as CoPs across RSV endpoints due to potential type I error inflation.

Overall, this study contributes a comprehensive correlates analysis of a large phase 3 efficacy trial of the mRNA-1345 RSV vaccine, supporting that RSV nAb and preF IgG bAb markers are CoPs for this vaccine. This work provides an opportunity to establish CoPs for RSV vaccines, paving the pathway for further evaluating whether such CoPs could also be determined for other RSV vaccine platforms.

## Methods

The protocol was approved by an institutional review board (Advarra), and the trial is being conducted according to the principles of the International Council for Harmonisation Technical Requirements for Registration of Pharmaceuticals for Human Use, E6(R2) Good Clinical Practice guidelines, the Declaration of Helsinki, and all national, state, and local laws or regulations. Prior to being enrolled in the study, all participants provided written informed consent.

### Study design

This study was designed to evaluate the immune response antibody markers, including the Day 29 antibody marker, baseline antibody marker, and fold-rise antibody marker (Day 29 marker/baseline marker), as CoRs and CoPs against each RSV endpoint in the phase 3 efficacy trial of mRNA-1345. Day 29 antibody markers were the primary interest in the analysis, which were extensively evaluated as CoRs and CoPs using different methods, including a univariable Cox regression model and mediation analysis. Both baseline and fold-rise antibody markers were also investigated as CoRs using a univariable Cox regression model.

From November 17, 2021, until December 23, 2022, 36,557 participants aged ≥60 years were randomly assigned in a 1:1 ratio to receive a single injection of mRNA-1345 50 μg or placebo at Day 1 (NCT05127434). Details of the double-blind study design, participant inclusion/exclusion criteria, and definitions were previously reported.[15] Serum samples were collected on Day 1 and Day 29 visits for all study participants, and a subset of study participants and all RSV cases by a case-cohort sampling design were assessed for immune response antibody markers, including RSV-A nAb, RSV-B nAb, preF IgG, and postF IgG. The immune correlate analyses were conducted to

study the immune response antibody markers against the 2 co-primary (RSV-LRTD-2+ and RSV-LRTD-3+) and the key secondary (RSV-ARD) clinical efficacy endpoints independently.

## Vaccine efficacy assessments

The 2 primary efficacy endpoints were prevention of a first episode of RSV-LRTD with ≥2 or ≥3 lower respiratory symptoms between 14 days and 12 months following injection. Key secondary efficacy endpoints included prevention of a first episode of RSV-ARD with ≥1 symptom and prevention of first hospitalization associated with RSV-ARD or RSV-LRTD between 14 days and 12 months after injection. Secondary endpoints included efficacy to prevent a first episode of RSV-LRTD or RSV-ARD by RSV subtype (RSV-A and RSV-B). RSV-LRTD was defined as RSV infection confirmed by reverse transcription–polymerase chain reaction (RT-PCR) and new or worsening lower respiratory symptoms for ≥24 h or confirmed RSV infection with radiologic evidence of pneumonia. RSV-ARD was defined as RT-PCR–confirmed RSV infection and new or worsening of ≥1 respiratory symptom for ≥24 h[15].

## Serum nAb assay

nAb activity against RSV subtypes A and B was assessed using a validated RSV-A and B microneutralization assay (Cerba Research, Netherlands) to quantify RSV nAbs in serum samples. Briefly, a constant amount of RSV was mixed with serial dilutions of human sera. If RSV-specific nAbs were present, RSV would be neutralized, and virus propagation in HEp-2 cells would be inhibited. Following an incubation period, cells were fixed and immuno-stained with a murine monoclonal antibody (Millipore Sigma, catalog number MAB858-1, clone 133-1H) directed against RSV F protein, followed by horseradish peroxidase-conjugated goat-anti-mouse antibody (Thermo Fisher Scientific, catalog number A16072, polyclonal antibody) and TrueBlue™ substrate. The plates were scanned with a UV Analyzer, and spot counts/well at each serum/antibody concentration were quantified. These values were used to determine the dilution of serum antibody at the 50% reduction point. Results were expressed as IU/mL as per the international standard antiserum to RSV-A and RSV-B from the World Health Organization (National Institute for Biological Standards and Control; https://www.nibsc.org).

## Serum bAb assay

A validated multiplex binding assay based on Luminex® technology was employed to quantify total IgG responses against RSV preF and postF. In brief, diluted study serum samples were incubated with two distinct Luminex® MagPlex microspheres, each coated with either preF or postF antigens, for 35 min at room temperature (RT). The antigen-conjugated microsphere-antibody complex was then incubated with an R-Phycoerythrin-labeled F(Ab')₂ Goat-anti-Human IgG secondary antibody (Jackson ImmunoResearch, catalog number 109-116-098, polyclonal antibody) for 60 min at RT. Following the secondary incubation, samples were analyzed using the Bio-Plex® 200 instrument. The fluorescence signal of the Fcγ fragment-specific secondary antibody was directly proportional to the concentration of preF and postF-specific serum IgG antibodies in the samples. A standard curve, assigned a value of 40,000 Arbitrary Units per milliliter (AU/mL) for both preF and postF, was used for quantification of the preF and postF antigen concentrations present in the serum samples. The measured signal is directly proportional to the amount of serum IgG antibodies specific for RSV preF and postF present in the serum samples. Results were expressed in arbitrary units per mL (AU/mL).

## Statistical analysis

The additional analysis of vaccine efficacy included data after >90% of participants (93.9% of the safety set) had completed ≥6 months of study follow-up (data cutoff date of April 30, 2023). Vaccine efficacy was defined as 100% × (1 − hazard ratio [mRNA-1345 vs placebo]), and

the confidence interval was based on a stratified Cox proportional hazard model, with Efron's method of tie handling and with the vaccination group as a fixed effect, adjusting for stratification factors at randomization.

The efficacy analyses were performed in the per-protocol efficacy (PPE) analysis population, which included all randomly assigned participants who received the vaccine or placebo, completed ≥1 visit or surveillance 14 days after injection, and had no major protocol deviations affecting assessment of efficacy outcomes. Subgroup analyses to assess the consistency of primary and key secondary efficacy endpoints included age, frailty status, presence or absence of RSV-LRTD risk factors (e.g., COPD and/or CHF), and the presence of ≥1 or the absence of comorbidities of interest.

All immune correlate analyses were conducted based on the prespecified statistical analysis plan (see the Statistical Analysis Plan in the Supplementary Information).

For each RSV endpoint, a synthetic baseline risk score was established by the logit of predictive probability of occurrence of an RSV endpoint using the ensemble super learner model to fit the occurrence of the RSV endpoint with 17 pre-specified baseline characteristic covariates (Figs. S37–S39; Tables S15–S18; see Supplementary Information) in placebo recipients only. Briefly, 7 learners in different types (linear, non-linear, nonparametric, ensemble learning) were trained in placebo recipients only using 2 levels of cross-validations (5-fold inner cross-validation, plus 5-fold outer cross-validation) to fit the binary outcome of each RSV endpoint, with either all or screened eligible baseline covariates, and then the fitted super learner model was chosen to predict the probability of occurrence of each RSV endpoint in vaccine recipients. By the classification accuracy metric of cross-validated area under the receiver operating characteristic (ROC) curve (CV-AUC), the baseline characteristics had mild prediction performance on placebo (CV-AUC: 0.632 for RSV-LRTD-2+, 0.644 for RSV-LRTD-3+, and 0.615 for RSV-ARD) and vaccine arms (CV-AUC: 0.630 for RSV-LRTD-2+, 0.670 for RSV-LRTD-3+, and 0.651 for RSV-ARD). Each categorical and quantitative variable was standardized to have a mean of 0 and a standard deviation of 1, and the missing values were imputed using the K-nearest neighbors method using the impute R package. The baseline risk score was standardized to have a mean of 0 and a standard deviation of 1 in the correlates analyses.

The case-cohort sampling design was implemented to (1) randomly select an immunogenicity subcohort from the full analysis set (see the Immunogenicity Subset Sampling Plan in the Supplementary Information) based on 3 baseline characteristics: age (60–74 vs. ≥75 years), LRTD risk status (present vs. absent), and region (Northern Hemisphere vs. Southern Hemisphere) and (2) select all postinjection RSV-ARD cases (between day 1 up to 24 months after injection). Participants were categorized as LRTD risk present if they had ≥1 congestive heart failure or chronic obstructive pulmonary disease risk factor at screening. Details for regions are shown in the Supplementary Information. IPS weighting was calculated based on the 2-phase sampling structure (see Supplementary Information), and all correlation analyses were adjusted by IPS weighting.

A univariable IPS-weight Cox proportional hazards regression model was utilized to evaluate the Day 29 and baseline antibody markers as CoRs against each RSV endpoint in both vaccine and placebo recipients. A univariable Cox regression model was also applied to fold-rise antibody markers only in vaccine recipients. For the Day 29 antibody marker, the point estimates and 95% CIs of covariate-adjusted HRs of each RSV endpoint (per 10-fold increase in the quantitative marker or per each vaccine tertile subgroup vs. placebo) were estimated, and nominal 2-sided p-values (Wald test) and FWER-adjusted p-values were reported, where the adjusted covariates included the actual stratification factors age group (60–74 vs. ≥75 years), LRTD risk status, and baseline risk score. The univariable IPS-weighted Cox regression model was used to conduct the mediation

analysis and estimate the cumulative incidence of each RSV endpoint through the study period (345 days after the Day 29 visit) per vaccine tertile subgroup, across a range of marker values by study period, and above a range of marker values by study period, respectively. The nominal 2-sided *p*-values and FWER-adjusted *p*-values for the interaction effect between treatment group and individual Day 29 markers by univariable IPS-weighted Cox regression full models (including the interaction term between treatment and marker) were also shown. Additionally, the covariate-adjusted HRs of each RSV endpoint (point estimates, 95% CIs, nominal *p*-values, FWER-adjusted *p*-values) per 10-fold increase in the quantitative baseline marker and per a standard deviation increment in fold-rise antibody marker were reported, respectively. Details are shown in the Supplementary Information.

### Reporting summary

Further information on research design is available in the Nature Portfolio Reporting Summary linked to this article.

### Data availability

As the trial is ongoing, access to patient-level data presented in this article and supporting clinical documents by qualified external researchers who provide methodologically sound scientific proposals may be available upon reasonable request for products or indications that have been approved by regulators in the relevant markets, and subject to review from 24 months after study completion. Such requests can be made to Moderna Inc., 325 Binney Street, Cambridge, MA 02142 «data_sharing@modernatx.com». A materials transfer and/or data access agreement with the sponsor will be required for accessing shared data. All other relevant data are presented in or provided with the paper. The figshare link for source data is: https://doi.org/10.6084/m9.figshare.29119217. The protocol is available online: Wilson et al.[15].

### Code availability

All analyses were done reproducibly based on R scripts in R version 4.2.2 that are provided as Supplementary Software (Supplementary Code 1).

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

## Acknowledgements

This work was funded by Moderna, Inc. Funding to pay the Open Access publication charges for this article was provided by Moderna, Inc. Moderna, Inc. contributed to the study design, clinical data collection, immune correlate analysis study design, immunogenicity data collection, statistical analysis, and manuscript writing. We would like to acknowledge the framework for statistics provided by: US Government-supported Coronavirus Prevention Network biostatistics team (Youyi Fong, David Benkeser, Nima Hejazi, Marco Carone, Avi Kenny, Lars van der Laan, Dean Follmann, Peter Gilbert). We would further like to acknowledge Professor Peter B. Gilbert for providing tremendously valuable review comments and suggestions. Finally, we would like to acknowledge the CONQUER RSV investigators and study participants. Editorial assistance was provided by MEDiSTRAVA in accordance with Good Publication Practice (GPP 2022) guidelines, funded by Moderna, Inc., and under the direction of the authors. C.J.A.D. is supported by a Clinician Scientist Fellowship from the UK Medical Research Council (MR/X001598/1).

## Author contributions

Conceptualization: C.M., L.L., A.K., S.K.S., C.A.S., L.Z., R.D., and H.Z. Methodology: C.M., S.K.S., L.Z., and H.Z. Data collection: C.M., G.J., G.P.M., A.K., C.J.A.D., N.L.C., N.L., F.P., S.G., J.G., and R.D. Data curation: C.M., L.L., J.D., and S.K.S. Visualization: C.M. and L.Z. Project administration: L.Z., H.Z., and C.J.A.D. Supervision: L.Z. and H.Z. Software: C.M. Study conduct and data interpretation: C.M., E.W., J.G., R.D., H.Z., J.D., L.L., S.K.S., C.A.S., S.G., and L.Z. Writing – original draft: C.M. Writing – review & editing: All coauthors

## Competing interests

C.M., J.D., L.L., A.K., N.L.C., N.L., F.P., S.G., S.K.S., C.A.S., J.G., E.W., R.D., H.Z., and L.Z. are employees of Moderna, Inc., and may hold stock/stock options in the company. C.J.A.D. acts on behalf of Newcastle upon Tyne Hospitals NHS Foundation Trust as an investigator on clinical trials sponsored by manufacturers of vaccines and antimicrobials, including Moderna, AstraZeneca, Synairgen, Janssen, and Valneva, and has received no personal financial payment for this work. C.J.A.D. has provided consultative advice to Synairgen on behalf of Newcastle University. G.P.M. has received a research grant from Merck and conducts clinical trials for Pfizer, Sanofi, and Moderna, Inc. G.J. has no conflicts to disclose.
