## [Peer Review file · Nature Communications]

Immune Correlates Analysis of mRNA-1345 RSV Vaccine Efficacy Clinical Trial

Corresponding Author: Dr Lingyi Zheng

Version 0:

Reviewer comments:

Reviewer #1

(Remarks to the Author)

This paper evaluates immune correlates for mRNA-1345, a vaccine designed to prevent RSV disease. Four different antibody assays are evaluated. The trial and data analysis are top-notch. I have minor comments.

Figure S23 could be clarified. I think you fit models with hazards $\lambda_{0(t)} \exp(Z A1)$ and $\lambda_{0(t)} \exp(Z B1 + Ab B2)$ with the estimate of A1 on the left panel and the estimate of B2 on the right?

Line 201. You write that the Prentice criteria is met for nAb and preF antibody biomarkers. Can you provide details. I think this would just mean reporting the interaction term for the model $\lambda_{0(t)} \exp(Z C1 + Ab C2 + Z Ab C3)$.

I'd not seen a mediation decomposition like you have done, but I like it. There are two different ways to decompose the overall effect into direct and indirect effects and so you take the geometric mean of the two.

The baseline CoR analysis seems incomplete as the vaccine baseline values should be superseded by the Abs induced by vaccination. Could you run the baseline CoR separately by vaccine and placebo groups and then discuss. For example, in the vaccine group you could run models for baseline Ab alone, D29 Ab alone, and the combination baseline Ab + D29 Ab to see to what extent D29 supersedes the baseline values in the vaccine group.

Line 280 you write "For each antibody marker, fold-rise in vaccine recipients decreased as the baseline marker level increased. Specifically, fold-rise of RSV-A nAb in breakthrough cases versus non-cases in vaccinees for each RSV endpoint was significantly higher." These are two different points yet the term "specifically" suggests the second sentence is details about the first.

Line 302 (and elsewhere) perhaps use abrogated rather than neutralized, since neutralization mostly means the assay in the rest of the paper.

Line 310, you suggest postF IgG is not a surrogate. Did you have specific analyses that showed this earlier?

Reviewer #2

(Remarks to the Author)

This manuscript is a comprehensive immune correlate analysis of a very large trial of the mRNA RSV vaccine, mRNA-1345, administered to older adults that demonstrated protective efficacy. The goal of the analysis is to determine if one or more of the antibody markers evaluated are correlated to protection or risk. The analysis is similar to analysis done for the COVID-19 mRNA vaccine trial.

This is an important study with valuable findings and brings forward several important concepts.

Critique

Overall, the paper is written in a style that makes it less accessible for many of the readers in the intended audience. Some

examples of that style that could be improved are provided below:

- The manuscript language includes a lot of statistical jargon that, although precise, will be unfamiliar to many readers. Many of these terms may be necessary but, when possible, it would be preferred to simplify.
- There are several sections of text with lists of numbers in the narrative, which are often also provided in a table. Lists of numbers in text are hard to read and comprehend. An example is found in lines 136-145. There the manuscript lists the geometric mean titers with confidence intervals for all four of the antibody markers for cases and non-cases. These numbers are all in Table 1. The reader might comprehend the text better if they are referred to the table, guided to the comparisons that the authors want to highlight rather than writing all the numbers in the text. The authors are advised to look at other sections of the narrative to identify lists of numbers and consider referring to tables where these numbers are found and summarize the point being made rather than the numbers. Generally, the authors should consider how to make their results section more concise and more pointed to emphasize the differences and the correlations but use the tables to provide the numbers.
- There are many places where the language could be more concise. For example, line 203, the statement “RSV endpoints (RSV-LRTD+2, RSV-LRTD+3, and RSV-ARD)” could be shortened to “all the RSV endpoints” since the reader is already familiar with the three endpoints that have already been defined earlier in the results section.
- Another example is the repeated use of “345 days after Day 29.” Please consider defining this with a more concise term (perhaps “study end” or “last observation” or something that the authors determine is accurate but shorter) and using that term throughout the manuscript, including figures.
- Consider shortening “RSV pre-F IgG bAb” to “pre-F IgG”
- Whenever possible, if this type of repeated terminology can be shortened to an easily absorbed label, it will make the manuscript more readable.

Figure 3 shows the correlation between the antibody markers on Day 29. One would be surprised if there was not a correlation between these responses and hence, it seems that seeing it in Figure 3 is of less interest to the reader than perhaps some of the other figures in the Supplement. Or if the authors believe that this correlation has important implications, this should be commented on the results or discussion. Do the authors think this correlation implies that one of these markers, for example pre-F IgG, could be used in isolation. If this is case, it would benefit the reader for this to be discussed and would justify inclusion of the current Figure 3, rather than replacing that figure with one that is providing further evidence of the antibody markers as correlates of protection.

Line 182, again, with the intent to make the manuscript more concise, the statement “There were 116 RSV-LRTD-2+ cases in the placebo recipients” could be deleted. The number is provided in Figure 4. Further, the number of cases among the placebo recipients, is not interpretable relative to the number in the vaccinees without providing the numerator, since the number of placebo recipients far exceed the number in each tertile.

One of the important findings of the study is that post-F IgG does not perform well as a surrogate marker. But there seems to be no figure(s) that show this included in the manuscript (though there are in the supplement). Since this is an important finding, it is worth showing that data in the main manuscript figures. The authors might consider including in Figure 4, the “panel d” from Figure S22 that shows that HR’s are the same across the tertiles for the post-F IgG which contrasts with the association of the lower HR with higher tertile for the other antibody markers.

Figure 5 is an important figure providing information about the antibody titer at which there will be only a lower probability of RSV illness. However, the text describing this figure (lines 207-214) seems like a “run-on” sentence and does not bring home the point that is being made with the figure.

The finding of the positive correlation of the fold-increase with RSV illness is of interest as this might not be intuitive. The authors have a good explanation for this, regarding the smaller fold-increase among vaccinees with a higher baseline. This finding may warrant more consideration in the Discussion since a particular fold-increase in titer is commonly used to define a vaccine response and might be assumed to confer protection. However, this study suggests that the absolute titer is more relevant to protection than how much the titer increased.

In the Discussion, the authors should comment on the overlap in the confidence intervals for the geometric mean titers between cases and non-cases (as shown in Figure 1). As well as the finding that many vaccinee cases had higher titers than placebo cases and non-cases. And finally, whether there is an antibody titer cutoff associated with a high level of protection that may serve as a target titer for vaccines in development that would be expected to confer protection.

The discussion could acknowledge that this vaccine only presents RSV F. And as such, only antibodies to RSV F can be induced and therefore studied as correlates of protection. Other vaccine approaches (e.g., live attenuated vaccines, vaccines presenting more than one antigen) may generate additional responses that might also be correlates of protection.

In the methods, the description of the assay technique for the Pre-F IgG should have a citation for the validation and/or to a more detailed description of the methods. In this manuscript, there should be information in the methods about how the detected antibodies are known to be IgG and not total of IgG, IgA, IgM.

Version 1:

Reviewer comments:

Reviewer #1

(Remarks to the Author)
no additional comments

Reviewer #2

(Remarks to the Author)

This revised manuscript is a comprehensive immune correlate analysis of a very large trial of the mRNA RSV vaccine, mRNA-1345, administered to older adults that demonstrated protective efficacy. The goal of the analysis is to determine if one or more of the antibody markers evaluated are correlated to protection or risk. The analysis is similar to analysis done for the COVID-19 mRNA vaccine trial.

This is an important study with valuable findings and brings forward several important concepts. The revision has improved readability and streamlined the narrative.

Reviewer 1

This paper evaluates immune correlates for mRNA-1345, a vaccine designed to prevent RSV disease. Four different antibody assays are evaluated. The trial and data analysis are top-notch. I have minor comments.

1. Figure S23 could be clarified. I think you fit models with hazards $\lambda_0(t) \exp(Z A1)$ and $\lambda_0(t) \exp(Z B1 + Ab B2)$ with the estimate of A1 on the left panel and the estimate of B2 on the right?

Response: The Reviewer is correct. In each panel of Figure S23, each row represents one model. For example, the first row in panel a represents the marginal treatment effect in the Cox regression model (without Day 29 Marker), ie, $RSV-LRTD-2+ \sim b1 * treatment$. The below rows represent the conditional treatment effect with a specific Day 29 antibody marker in the Cox regression model, ie, $RSV-LRTD-2+ \sim b1 * treatment + b2 * Day\ 29\ antibody\ marker$. Hence, the left side of the Figure is the treatment effect (b1) of each model, and the right side is the Day 29 Marker effect (b2). Similar interpretations were applied to panels (b) and (c). Please note that all Cox regression models were adjusted by baseline risk factors, fitted with the inverse probability of sampling weights. We have accordingly clarified the legend of Figure S23 to address this comment.

Revised text: *In each panel, the rows highlighted in yellow represent the marginal treatment (vaccine) effect, with the below rows representing the conditional treatment effect with a specific Day 29 marker. Rows highlighted in blue represent the top-performance CoRs/CoPs. Hazard ratios for the treatment (vaccine) effect of each model are shown on the left, while hazard ratios for the Day 29 marker effect are shown on the right. Baseline risk factors were adjusted in the univariable inverse probability of sampling weighted Cox regression model, including the actual stratification factors age and LRTD at-risk, and baseline risk score.*

2. Line 201. You write that the Prentice criteria is met for nAb and preF antibody biomarkers. Can you provide details. I think this would just mean reporting the interaction term for the model $\lambda_0(t) \exp(Z C1 + Ab C2 + Z Ab C3)$.

Response: We thank the Reviewer for their interest. The Prentice surrogate endpoint criteria for an antibody marker include:

- 1) Significant treatment effect (without including the antibody marker)
- 2) Significant association between the antibody marker and the treatment
- 3) Treatment effect conditional on the antibody marker is not significant and should be close to zero
- 4) No significant interaction effect between the antibody marker and the treatment

In Figure S23, the marginal vaccine (treatment) effect for each RSV endpoint is statistically significant (first row of panels a-c), meeting Prentice criterion: (1); the association between the nAb or preF IgG antibody markers and an RSV endpoint is significant (or strongly associated) based on the antibody marker effect represented in rows 2-4 for all panels on the right side of the Figure, aligning with Prentice criterion (2); the vaccine (treatment) effect conditional on nAb or preF IgG antibody markers was no longer significant compared to the marginal vaccine effect (without an antibody marker) based on the treatment effect in rows 2-4 for all panels on the left side of the Figure, satisfying Prentice criterion (3). Additionally, based on the Table S9, all the interaction effects between nAb or preF IgG antibody markers and an RSV endpoint are not significant, satisfying Prentice criterion (4). Overall, the data and statistical analysis supports nAb and preF IgG antibodies, satisfying the Prentice criteria of surrogate markers. We note that these details are provided in the Discussion, which we have slightly revised for clarity.

Revised text: *Our comprehensive immune correlates analysis of the phase 3 efficacy trial of mRNA-1345 versus placebo demonstrated that RSV nAb (A and B subtypes) and RSV preF IgG bAb markers assessed at Day 29 were significantly inversely correlated with primary and key secondary endpoints (RSV-LRTD-2+ and RSV-ARD) and were also strongly and consistently inversely correlated with another primary endpoint (RSV-LRTD-3+). The treatment effect for each RSV endpoint conditional on Day 29 RSV-A nAb, RSV-B nAb, and preF IgG markers was not significant and was nearly mediated by these markers (estimated HR close to 1). The interaction effect between treatment and these antibody markers was also not significant, supporting that RSV risk only depends on antibody level, regardless if the antibody is elicited by vaccine or natural RSV infection. Therefore, by the Prentice criteria, RSV nAb and preF IgG markers are supported as surrogate endpoints/CoPs and can be considered as surrogate markers for RSV endpoints.*

3. I'd not seen a mediation decomposition like you have done, but I like it. There are two different ways to decompose the overall effect into direct and indirect effects and so you take the geometric mean of the two.

Response: We sincerely thank the Reviewer for their feedback. Generally, the total effect can be decomposed into "controlled direct effect × controlled indirect effect" or "natural direct effect × natural indirect effect" under the Cox regression model, and indeed the natural decomposition is commonly used. However, in this pivotal mRNA-1345 RSV vaccine study, all participants were RSV baseline positive, and the Day 29 antibody marker levels for the placebo group overlapped with those in the vaccine group, which satisfied the positivity assumption (of note, the average antibody level in the vaccine group was approximately 10 times higher). Hence, the placebo is considered as a natural "vaccine" treatment, and the vaccine group and the placebo group can serve as a control for each other. Additionally, the total effect was decomposed into the indirect and direct effect by two different methods (ie, natural decomposition and controlled decomposition), and obtained the geometric mean for each of the indirect and direct effect. Thus, this adapted indirect effect can be used to interpret the extent at which the antibody marker can mediate the vaccine efficacy.

4. The baseline CoR analysis seems incomplete as the vaccine baseline values should be superseded by the Abs induced by vaccination. Could you run the baseline CoR separately by vaccine and placebo groups and then discuss. For example, in the vaccine group you could run models for baseline Ab alone, D29 Ab alone, and the combination baseline Ab + D29 Ab to see to what extent D29 supersedes the baseline values in the vaccine group.

Response: To clarify, the correlate analysis plan prespecified for assessments on Day 1 (baseline) antibody markers as CoRs. Figure S32 displays the Cox-based analysis results for each of the 4 baseline antibody markers against each RSV endpoint in both vaccine and placebo groups. Per the Reviewer's interest, we conducted the corresponding analysis in the vaccine group and evaluated the CoRs for each individual Day 1 antibody marker, each individual Day 29 antibody marker, as well as the combined Day 1 and Day 29 antibody markers for each assay, respectively. In Figure 1 below, the Cox-based results show that the individual Day 1 markers

demonstrated stronger evidence (smaller HRs and smaller p-values) for correlating with the RSV endpoints compared to the same individual Day 29 markers in the univariable (antibody) Cox regression model in the vaccine group. This result could be explained by the baseline antibody titers (or levels) representing pre-existing immunity (eg prior RSV infection), so that higher baseline antibody titers (or levels) lead to more robust immune response after vaccination. However, such strong statistical association between baseline antibody markers and RSV endpoints in the vaccine group is overestimated compared to that observed in the placebo group (**Figure 2** below, panel a – c), due to potential confounding with postvaccination immune response (high correlations between baseline and Day 29 antibody markers with Spearman rank correlation > 0.5). Importantly, the post-vaccination elicited antibody response (Day 29 antibody marker) has more fundamental biological relevance with protection from RSV disease than pre-existing immunity (baseline antibody) in the vaccine group. Overall, the baseline antibody markers are not reasonably likely to predict vaccine efficacy, as they are not able to mediate any vaccine efficacy (as shown in Figure S32). Additionally, the association between Day 29 antibody markers and RSV endpoints (below, panel d – f) in the vaccine group is generally comparable to that in the placebo group (**Figure 2** below, panel d – f). This is consistent with the non-significant interaction effect between Day 29 antibody markers and the vaccine group (shown in Table S9).

Alternately, when the same Day 1 and Day 29 antibody markers were included in the model, only the Day 1 antibody marker is significant, while none of the Day 29 antibody markers are significant for any RSV endpoint. The high correlations between the Day 1 and Day 29 antibody marker tends to inversely overestimate the effects (ie, Day 1 marker effect is negatively overestimated and the Day 29 marker effect is positively overestimated). Technically, this phenomenon could occur because the internal Newton-Raphson optimization algorithm may not converge due to the high-correlations.

Figure 1. Forest plot of univariable IPS-weighted Cox regression model summary by (1) fitting each RSV endpoint with an individual Day 1 antibody marker in the vaccine group shown in panel (a-c); (2) fitting each RSV endpoint with an individual Day 29 antibody marker in the vaccine group shown in panel (d-f); (3) fitting each RSV endpoint with an individual Day 1 and an individual Day 29 antibody marker by the same assay type shown in panel (g-i), respectively. Baseline risk factors were adjusted in the Cox regression model, including the actual stratification factors age and LRTD at-risk, and baseline risk score.

Figure 2 below displays the Cox-based analysis results evaluating individual Day 1 (baseline) and Day 29 antibody markers as CoRs against each RSV endpoint in the placebo group. Since the Day 1 and Day 29 antibody marker levels in the placebo group are highly correlated, the CoR results for Day 1 antibody markers in **Figure 2** panel (a-c) are very similar, as are those for Day 29 antibody markers in **Figure 2** panel (d-f). Due to the multicollinearity between the Day 1 and Day 29 antibody markers in the placebo group (Spearman rank correlations approximate to 0.9), the

Cox model including both Day 1 and Day 29 antibody markers for the placebo group did not converge and hence the corresponding results were not reported here.

Figure 2. Forest plot of univariable IPS-weighted Cox regression model summary by (1) fitting each RSV endpoint with an individual Day 1 antibody marker in the placebo group shown in panel (a-c); (2) fitting each RSV endpoint with an individual Day 29 antibody marker in the placebo group shown in panel (d-f), respectively. Baseline risk factors were adjusted in the Cox regression model, including the actual stratification factors age and LRTD at-risk, and baseline risk score.

Overall, the nAb and preF IgG markers showed consistent evidence of being CoRs for each RSV endpoint in this exploratory analysis. Moreover, the preF IgG marker showed more robust correlations with RSV endpoints than the nAb markers, which could be due to the smaller standard deviation. Of note in the vaccine or placebo group, the correlation between some individual Day 29 antibody markers and the RSV-LRTD 3+ endpoint is not significant, which could be largely due to small number of cases in the vaccine group (n = 19) and in the placebo group (n = 49). While we appreciate the Reviewer's suggestion, no change has been made to the manuscript to address this comment.

- Line 280 you write "For each antibody marker, fold-rise in vaccine recipients decreased as the baseline marker level increased. Specifically, fold-rise of RSV-A nAb in breakthrough cases versus

non-cases in vaccinees for each RSV endpoint was significantly higher.” These are two different points yet the term “specifically” suggests the second sentence is details about the first.

Response: We agree with the Reviewer and have accordingly updated the text in line 280 for clarity.

Revised text: *For each antibody marker, fold-rise in vaccine recipients decreased as the baseline marker level increased; fold-rise of RSV-A nAb was significantly higher in breakthrough cases versus non-cases in vaccinees for each RSV endpoint.*

6. Line 302 (and elsewhere) perhaps use abrogated rather than neutralized, since neutralization mostly means the assay in the rest of the paper.

Response: The Results and Discussion sections have been accordingly updated to use the term mediated instead of neutralized.

7. Line 310, you suggest postF IgG is not a surrogate. Did you have specific analyses that showed this earlier?

Response: To clarify, the specific analyses that support this conclusion are already included in the manuscript. For each analysis in this manuscript, we evaluated 4 antibody markers (RSV-A nAb, RSV-B nAb, preF IgG, and postF IgG) as a CoR or CoP. Overall, postF IgG showed some limitations and lacked consistency during CoP assessments using the different methods, as noted in the first paragraph of the Discussion (please see further below for reference). Further, we refer the Reviewer to the Results section, lines 175–176, that states “... RSV risk decreased from placebo through low, medium, high vaccine tertile subgroups for each marker (except RSV postF IgG) (Figs. S17-S19)”. Secondly, in lines 187–189, the manuscript states “Compared to the trend of decreasing RSV-LRTD-2+ risk with increasing vaccine tertiles by RSV-A nAb, RSV-B nAb, and RSV preF IgG, there was no such evidence for RSV postF IgG”. These data support that postF IgG is not a predictive marker, hence we suggest that postF IgG is not a CoP.

Current text (Discussion): *Conversely, while Day 29 postF IgG levels also had strong inverse correlations with RSV endpoints, the treatment effect against RSV-LRTD-2+ conditional on postF IgG levels was somewhat closer to the marginal treatment effect. Importantly, evidence of being a CoR for Day 29 postF IgG under quantitative and qualitative univariable Cox regression models was not consistent, indicating that postF IgG is not a CoP. The complementary analysis of postF IgG echoed the historical lessons of the failure of postF-based RSV vaccines and further supports the development strategy of mRNA-1345, which successfully protects against RSV disease by targeting RSV prefusion F protein^{22, 23}.*

Reviewer 2

This manuscript is a comprehensive immune correlate analysis of a very large trial of the mRNA RSV vaccine, mRNA-1345, administered to older adults that demonstrated protective efficacy. The goal of the analysis is to determine if one or more of the antibody markers evaluated are correlated to protection or risk. The analysis is similar to analysis done for the COVID-19 mRNA vaccine trial.

This is an important study with valuable findings and brings forward several important concepts.

Critique

Overall, the paper is written in a style that makes it less accessible for many of the readers in the intended audience. Some examples of that style that could be improved are provided below:

1. The manuscript language includes a lot of statistical jargon that, although precise, will be unfamiliar to many readers. Many of these terms may be necessary but, when possible, it would be preferred to simplify.

Response: We thank the Reviewer for noting this is an important study with valuable findings. We have accordingly revised and simplified the manuscript text based on this comment and the

below comments. As noted by the Reviewer, certain statistical terms have been retained in the manuscript to ensure that readers are informed about the statistical analyses used in the study.

2. There are several sections of text with lists of numbers in the narrative, which are often also provided in a table. Lists of numbers in text are hard to read and comprehend. An example is found in lines 136-145. There the manuscript lists the geometric mean titers with confidence intervals for all four of the antibody markers for cases and non-cases. These numbers are all in Table 1. The reader might comprehend the text better if they are referred to the table, guided to the comparisons that the authors want to highlight rather than writing all the numbers in the text. The authors are advised to look at other sections of the narrative to identify lists of numbers and consider referring to tables where these numbers are found and summarize the point being made rather than the numbers. Generally, the authors should consider how to make their results section more concise and more pointed to emphasize the differences and the correlations but use the tables to provide the numbers.

Response: As requested, the manuscript has been revised to refer readers to specific Tables and Figures where the exact values are represented, with the aim to more concisely and narratively describe the findings.

3. There are many places where the language could be more concise. For example, line 203, the statement “RSV endpoints (RSV-LRTD+2, RSV-LRTD+3, and RSV-ARD)” could be shortened to “all the RSV endpoints” since the reader is already familiar with the three endpoints that have already been defined earlier in the results section.

Response: The manuscript has been amended at line 203 and in other instances (as applicable) to accordingly state “all RSV endpoints” instead of “RSV-LRTD+2, RSV-LRTD+3, and RSV-ARD.”

4. Another example is the repeated use of “345 days after Day 29.” Please consider defining this with a more concise term (perhaps “study end” or “last observation” or something that the

authors determine is accurate but shorter) and using that term throughout the manuscript, including figures.

Response: The manuscript and supplement, including the respective Figures, have been revised accordingly, with “345 days after Day 29 visit” changed to “study period”.

5. Consider shortening “RSV pre-F IgG bAb” to “pre-F IgG”

Response: The manuscript has been revised as suggested, with RSV Pre-F IgG bAb changed to preF IgG.

Revised text: *The objective of this analysis was to evaluate nAb against RSV-A and -B subtypes, as well as IgG bAb to RSV preF or postfusion (postF), as (1) measured on Day 29 (hereafter, “Day 29 antibody marker”); (2) measured on Day 1 (day of receiving injection; hereafter, “baseline antibody marker”); (3) fold-rise from baseline antibody to Day 29 antibody marker (hereafter, “fold-rise antibody marker”), as correlates of risk (CoRs) and CoPs against primary and key secondary VE endpoints (hereafter, “RSV endpoints”).*

6. Whenever possible, if this type of repeated terminology can be shortened to an easily absorbed label, it will make the manuscript more readable.

Response: Based on this and the above comment, any repeated terminology has been accordingly further abbreviated to enhance readability.

7. Figure 3 shows the correlation between the antibody markers on Day 29. One would be surprised if there was not a correlation between these responses and hence, it seems that seeing it in Figure 3 is of less interest to the reader than perhaps some of the other figures in the Supplement. Or if the authors believe that this correlation has important implications, this should be commented on the results or discussion. Do the authors think this correlation implies that one of these markers, for example pre-F IgG, could be used in isolation. If this is case, it

would benefit the reader for this to be discussed and would justify inclusion of the current Figure 3, rather than replacing that figure with one that is providing further evidence of the antibody markers as correlates of protection.

Response: Figure 3 was included in the main text of the manuscript to support demonstration of the high correlations between the Day 29 antibody markers. Our intention was to allow readers to readily perceive how the vaccine- or placebo-elicited antibody markers correlate with each other, prior to describing the specific analysis. Based on the Reviewer's feedback and since Figure 3 was largely similar to Figure S10 (now Figure S7, to reflect its now earlier citation position in the manuscript), we have removed Figure 3 and solely cite Figure S7 to highlight these findings.

8. Line 182, again, with the intent to make the manuscript more concise, the statement "There were 116 RSV-LRTD-2+ cases in the placebo recipients" could be deleted. The number is provided in Figure 4. Further, the number of cases among the placebo recipients, is not interpretable relative to the number in the vaccinees without providing the numerator, since the number of placebo recipients far exceed the number in each tertile.

Response: This statement has been accordingly removed from the Results section of the manuscript.

9. One of the important findings of the study is that post-F IgG does not perform well as a surrogate marker. But there seems to be no figure(s) that show this included in the manuscript (though there are in the supplement). Since this is an important finding, it is worth showing that data in the main manuscript figures. The authors might consider including in Figure 4, the "panel d" from Figure S22 that shows that HR's are the same across the tertiles for the post-F IgG which contrasts with the association of the lower HR with higher tertile for the other antibody markers.

Response: We thank the Reviewer for their insightful comment. However, as postF IgG is not a predictive marker, our preference is to retain this finding within the supplement so as to have

the main manuscript highlight the key findings from this analysis: that RSV nAb and preF IgG antibody markers are considered surrogate markers for RSV endpoints. Further, the mRNA-1345 vaccine was developed based on the RSV-A2 strain protein sequence, stabilized in the preF conformation. Apart from nAbs, preF IgG is thus of key interest in the evaluation as a CoR and CoP for RSV endpoints. The finding that postF IgG was not a good surrogate marker is important but not novel, as a previous phase 2b study reported that postF IgG does not protect against RSV disease in older adults (Falloon, et al. 2017).

10. Figure 5 is an important figure providing information about the antibody titer at which there will be only a lower probability of RSV illness. However, the text describing this figure (lines 207-214) seems like a “run-on” sentence and does not bring home the point that is being made with the figure.

Response: The description of the findings presented in Figure 5 (now Figure 4; see response to Comment 7) has been accordingly revised to provide a high-level summary of the results presented in Figure 5.

Revised text: Fig. 4 displays further correlates analysis specifically for RSV-LRTD-2+ and shows the estimated (1) cumulative incidence of RSV-LRTD-2+ during the study period across a range of assigned marker levels by vaccine and placebo; (2) controlled VE against RSV-LRTD-2+ during the study period across a range of assigned marker levels; and (3) cumulative incidence of RSV-LRTD-2+ during the study period above a range of assigned marker levels (thresholds) by Day 29 RSV-A nAb and RSV preF IgG markers of vaccinees (see supplementary materials), respectively. Specifically, for both Day 29 RSV-A nAb and preF IgG, the estimated cumulative incidence of RSV-LRTD-2+ by vaccine and placebo group was similar and overlapped by bootstrap pointwise 95% CIs; estimated VE increased as the antibody marker level increased.

11. The finding of the positive correlation of the fold-increase with RSV illness is of interest as this might not be intuitive. The authors have a good explanation for this, regarding the smaller fold-increase among vaccinees with a higher baseline. This finding may warrant more consideration in the Discussion since a particular fold-increase in titer is commonly used to define a vaccine

response and might be assumed to confer protection. However, this study suggests that the absolute titer is more relevant to protection than how much the titer increased.

Response: We appreciate the Reviewer's interest and have expanded the Discussion section to provide further details on fold-rise in antibody levels in this study.

Revised text: *We also investigated baseline and fold-rise antibody markers as CoRs against RSV endpoints. These analyses found that all baseline antibody markers had strong inverse correlation with RSV endpoints, similar to Day 29 antibody markers; nevertheless, baseline antibody markers were less reliable in predicting VE compared to Day 29 markers because the corresponding conditional treatment effect was largely unchanged compared to the marginal treatment effect. Conversely, all fold-rise antibody markers were positively correlated with RSV endpoints, and correlate strengths between fold-rise markers and RSV endpoints were comparable to Day 29 markers. However, such positive correlation between fold-rise antibody markers and RSV endpoints was very likely reversely masked by baseline antibody markers due to high negative correlation between fold-rise and baseline antibody markers (Spearman correlation <-0.8). Fold-rise in antibody titers or levels is commonly used to evaluate vaccine-elicited immune responses and is often considered as a secondary endpoint in vaccine trials. However, in this study, the absolute antibody titer/level was more reasonably correlated with the protection for RSV disease as compared to the corresponding fold-rise antibody marker.*

12. In the Discussion, the authors should comment on the overlap in the confidence intervals for the geometric mean titers between cases and non-cases (as shown in Figure 1). As well as the finding that many vaccinee cases had higher titers than placebo cases and non-cases. And finally, whether there is an antibody titer cutoff associated with a high level of protection that may serve as a target titer for vaccines in development that would be expected to confer protection.

Response: The Discussion has been updated to describe and comment upon these findings, as requested.

Revised text: *The advantage of our method is to guarantee the monotone decreasing trend of the cumulative risk with marker threshold (similar to the method in Van der Laan et al.²⁷) and narrower CIs. However, it is notable that multiple cases in the vaccine group had high nAb titers or preF IgG levels (Fig. S4 – S6), indicating that the nAb or preF IgG marker may not be a mechanistic CoP, and that this subgroup with high nAb titer or preF IgG level cannot attain a very low level of RSV risk. Additionally, many non-cases in the vaccine group had relatively low nAb titers or preF IgG levels, indicating that other types of antibody markers may exist to protect against RSV disease. Future studies can evaluate additional antibody response generated by other vaccine approaches (eg live attenuated vaccines, or vaccines containing more than one antigen) as potential CoPs for RSV disease.*

Revised text: *We also investigated baseline and fold-rise antibody markers as CoRs against RSV endpoints. These analyses found that all baseline antibody markers had strong inverse correlation with RSV endpoints as Day 29 antibody markers, which coincided with the GMT (or GMC) ratio for baseline nAb titers (or preF IgG levels) for cases versus non-cases in the vaccine or placebo groups being lower than that observed for Day 29 antibody markers (Table S5 – S7). Of note, the ratio of cases versus non-cases by an RSV endpoint (eg RSV-LRTD-3+) for some antibody markers may cross-over one, which may result in large variability due to the relevant small number of non-cases.*

13. The discussion could acknowledge that this vaccine only presents RSV F. And as such, only antibodies to RSV F can be induced and therefore studied as correlates of protection. Other vaccine approaches (e.g., live attenuated vaccines, vaccines presenting more than one antigen) may generate additional responses that might also be correlates of protection.

Response: We note that the mediation analysis findings from this study have shown that the postvaccination (eg, Day 29) nAb and preF IgG markers can mediate vaccine efficacy reasonably well and thus can be supported as surrogate markers to predict clinical benefit. However, per the Reviewer's comment here and comment 12 above, the Discussion has been revised further to acknowledge that alternative vaccine approaches will need further evaluations.

Revised text: Future studies can evaluate additional antibody response generated by alternate vaccine approaches (eg live attenuated vaccines, or vaccines containing more than one antigen) as potential CoPs for RSV disease.

14. In the methods, the description of the assay technique for the Pre-F IgG should have a citation for the validation and/or to a more detailed description of the methods. In this manuscript, there should be information in the methods about how the detected antibodies are known to be IgG and not total of IgG, IgA, IgM.

Response: The methods section has been updated to provide a more detailed description of the multiplex binding assay used to measure RSV preF and postF IgG.

Revised text: A validated multiplex binding assay based on Luminex® technology, was employed to quantify total IgG responses against RSV preF and postF. In brief, diluted study serum samples were incubated with two distinct Luminex® MagPlex microspheres, each coated with either preF or postF antigens, for 35 minutes at room temperature (RT). The antigen-conjugated microsphere-antibody complex was then incubated with an R-Phycoerythrin-labeled F(Ab')₂ goat anti-Human IgG secondary antibody for 60 minutes at RT. Following the secondary incubation, samples were analyzed using the Bio-Plex® 200 instrument. The fluorescence signal of the Fcy fragment-specific secondary antibody was directly proportional to the concentration of preF and postF specific serum IgG antibodies in the samples. A standard curve, assigned a value of 40,000 Arbitrary Units per milliliter (AU/mL) for both preF and postF, was used for quantification of the preF and postF antigen concentrations present in the serum samples. The measured signal is directly proportional to the amount of serum IgG antibodies specific for RSV preF and postF present in the serum samples. Results were expressed in arbitrary units per mL (AU/mL).

Editorial

POLICIES AND FORMS REQUIRED FOR RESUBMISSION

1. Please complete or update the following checklist(s) to verify compliance with our research ethics and data reporting standards. Address all points on the checklist, revising your manuscript in response to the points if needed.

The form(s) must be downloaded and completed in Adobe Reader rather than opened in a web browser. Each form must be uploaded as a Related Manuscript file at the time of resubmission.

Editorial policy checklist:

<https://www.nature.com/documents/nr-editorial-policy-checklist.pdf>

Reporting summary:

Response: We confirm that that the Editorial policy checklist and Reporting summary have been completed and uploaded as related manuscript files during resubmission.

2. Nature journals have recently announced an update to our guidance on reporting on sex and gender in research studies (<https://www.nature.com/articles/s41467-022-30398-1>). We strongly encourage researchers to follow the Sex and Gender Equity in Research – SAGER – guidelines (<https://researchintegrityjournal.biomedcentral.com/articles/10.1186/s41073-016-0007-6>) and to include sex and gender considerations for studies involving humans, vertebrate animals and cell lines where relevant to the topic of study (an overview can be found here: <https://www.ease.org.uk/wp-content/uploads/2016/09/Sager.for-web.pdf>). Authors should use the terms sex (biological attribute) and gender (shaped by social and cultural circumstances) carefully in order to avoid confusing both terms.

When preparing your revised manuscript, please be aware of our guidance on Sex and Gender reporting (<https://www.nature.com/nature-portfolio/editorial-policies/ethics-and-biosecurity#ethics-policy>).

Please note that we require that the following recommendations from the guidelines are followed:

- i. If the research findings apply to only one sex or gender, that must be indicated in the title and/or abstract.

ii(a). For studies involving vertebrates animal and cell lines- The Reporting Summary should include whether sex was considered in the study design.

ii(b). For studies involving human research participants- The Reporting Summary should include whether sex and/or gender was considered in the study design and whether sex and/or gender of participants was determined based on self-report or assigned (and methodology used).

iii. Data should be reported disaggregated for sex and gender where this information has been collected and consent has been obtained for reporting and sharing individual-level data; disaggregated numbers for individual experiments must be provided in the source data as appropriate whereas overall numbers may be provided in the Nature Portfolio Reporting Summary.

Information on the points above should be included in the revised manuscript and detailed in the cover letter.

In addition, please note that if sex- and gender-based analyses have been performed a priori, results should be reported regardless of positive or negative outcome. We discourage conducting post hoc sex- and gender-based analysis if the study design is insufficient (for example, low sample size) to enable meaningful conclusions.

If no sex- and gender-based analyses have been performed, please indicate the reasons for the lack of these analyses in the Reporting Summary.

Response: We confirm that no reporting by sex or gender was performed in this study, which is indicated in the submitted Reporting Summary.

DATA AND CODE AVAILABILITY

3. All Nature Communications manuscripts must include a “Data Availability” section after the Methods section but before the References. If any of the data can only be shared on request or are subject to restrictions, please specify the reasons and explain how, when, and by whom the data can be accessed. For more information on this policy and a list of examples, see: <https://www.nature.com/documents/nr-data-availability-statements-data-citations.pdf>

Response: We confirm that the Data Availability section has been included in the manuscript after the Methods section but before the References section.

4. We strongly encourage you to deposit all new data associated with the paper in a persistent repository where they can be freely and enduringly accessed. We recommend submitting the data to discipline-specific and community-recognized repositories; a list of repositories is provided here: <http://www.nature.com/sdata/policies/repositories>
Refer to our data policies here: <https://www.nature.com/nature-portfolio/editorial-policies/reporting-standards#availability-of-data>

Response: As noted in our Data Availability statement, data associated with this study will be provided to requestors upon reasonable request for products or indications that have been approved by regulators in the relevant markets, and after review.

5. To maximise the reproducibility of research data, we strongly encourage you to provide a file containing the raw data underlying the following types of display items:
 - Any reported means/averages in box plots, bar charts, and tables
 - Dot plots/scatter plots, especially when there are overlapping points
 - Line graphs
 - Uncropped and unprocessed scans of all blots and gels including all quantified replicates. The edge of membranes, molecular weight ladders and loading controls should be presented on all blots. Where membranes have been cut, please ensure that at least one marker above and below is present. For an example of presentation of full scan blots, see the Source Data file of <https://www.nature.com/articles/s41467-020-16984-1#Sec35> and for more information, please refer to <https://www.nature.com/nature-research/editorial-policies/image-integrity>.

The data should be provided in a single Excel file with data for each figure/table in a separate sheet, or in multiple labelled files within a zipped folder. Name this file or folder 'Source Data', and include a brief description in your cover letter. The "Data Availability" section should also include the statement "Source data are provided with this paper."

To learn more about our motivation behind this policy, please see:

<https://www.nature.com/articles/s41467-018-06012-8>

A Source Data file is not necessary if all display items presented in the main manuscript and supplementary information can be reproduced from raw data and code that have already been shared in a public repository.

Response: As stated above and as noted in our Data Availability statement, data associated with this study will be provided to requestors upon reasonable request for products or indications that have been approved by regulators in the relevant markets, and after review.

ORCID

6. Nature Communications is committed to improving transparency in authorship. As part of our efforts in this direction, we are now requesting that all authors identified as 'corresponding author' create and link their Open Researcher and Contributor Identifier (ORCID) with their account on the Manuscript Tracking System prior to acceptance. ORCID helps the scientific community achieve unambiguous attribution of all scholarly contributions.

Response: We confirm that the Open Researcher and Contributor Identifier for the corresponding author has been linked on the Manuscript Tracking System.

References

1. Falloon, J., Yu, J., Esser, M., Villafana, T. Yu, L., Dubovsky, F. Takas, T., Levin, M.J., Falsey, A.R. An Adjuvanted, Postfusion F Protein-Based Vaccine Did Not Prevent Respiratory Syncytial Virus Illness in Older Adults. *J Infect Dis* 216:1362-1370 (2017).